# Relevant Serum Endoplasmic Reticulum Stress Biomarkers in Type 2 Diabetes and Its Complications: A Systematic Review and Meta-Analysis

**DOI:** 10.3390/antiox13121564

**Published:** 2024-12-19

**Authors:** José Rafael Villafan-Bernal, Francisco Barajas-Olmos, Iris Paola Guzmán-Guzmán, Angélica Martínez-Hernández, Cecilia Contreras-Cubas, Humberto García-Ortiz, Monserrat I. Morales-Rivera, Raigam Jafet Martínez-Portilla, Lorena Orozco

**Affiliations:** 1Immunogenomics and Metabolic Diseases Laboratory, Instituto Nacional de Medicina Genómica, SS, Mexico City 14610, Mexico; fbarajas@inmegen.gob.mx (F.B.-O.); amartinez@inmegen.gob.mx (A.M.-H.); ccontreras@inmegen.gob.mx (C.C.-C.); hgarcia@inmegen.gob.mx (H.G.-O.); mrivera@inmegen.edu.mx (M.I.M.-R.); 2Investigador por México, Consejo Nacional de Humanidades Ciencia y Tecnología (CONAHCYT), Mexico City 03940, Mexico; 3Iberoamerican Research Network in Translational, Molecular and Maternal-Fetal Medicine, Mexico City 01010, Mexico; raifet@hotmail.com; 4Laboratory of Multidisciplinary Research and Biomedical Innovation, Universidad Autónoma de Guerrero, Chilpancingo 39086, Guerrero, Mexico; pao_nkiller@yahoo.com.mx; 5Postdoctoral Researcher, Consejo Nacional de Humanidades Ciencias y Tecnologías, Mexico City 03940, Mexico

**Keywords:** endoplasmic reticulum stress, biomarkers, type 2 diabetes, diabetes complications, secretagogin, peroxiredoxins, HSP-70

## Abstract

Endoplasmic reticulum stress (ERS) is activated in all cells by stressors such as hyperglycemia. However, it remains unclear which specific serum biomarkers of ERS are consistently altered in type 2 diabetes (T2D). We aimed to identify serum ERS biomarkers that are consistently altered in T2D and its complications, and their correlation with metabolic and anthropometric variables. We performed a systematic review and meta-analysis following Preferred Reporting Items for Systematic Reviews and Meta-Analyses (PRISMA) and Meta-Analyses and Systematic Reviews of Observational Studies (MOOSE). The risk of bias was assessed using the Newcastle–Ottawa scale. Random-effects models weighted by the inverse variance were employed to estimate the standardized mean difference and correlations as effect size measures. Indicators of heterogeneity and meta-regressions were evaluated. Of the 1206 identified studies, 22 were finally included, representing 11,953 subjects (2224 with T2D and 9992 non-diabetic controls). Most studies were of high quality. Compared with controls, subjects with T2D had higher circulating levels of heat shock protein 70 (HSP70; SMD: 2.30, 95% CI 1.13–3.46; *p* < 0.001) and secretagogin (SMD: 0.60, 95%CI 0.19–1.01; *p* < 0.001). They also had higher serum levels of peroxiredoxin-1, -2, -4, and -6. Secretagogin inversely correlated with HOMA-IR, yet positively correlated with HOMA-B, HbA1c, and FPG. PRX4 negatively correlated with HbA1c and FPG, while HSP70 positively correlated with HbA1c. In conclusion, six ERS biomarkers are consistently elevated in human T2D and correlate with glycemic control, insulin resistance, and β-cell function. Emerging evidence links serum ERS biomarkers to diabetes complications, but further research should evaluate their prognostic implications.

## 1. Introduction

Endoplasmic reticulum stress (ERS) is initially a physiological adaptative process, but its perpetual activation can be deleterious for cell, tissue, and organ homeostasis [1]. ERS is triggered by various stimuli, including oxidative stress, reactive oxygen species (ROS), free fatty acids, hyperglycemia, hyperinsulinemia, high fructose intake, high-fat diets, obesity, and nutritional excess [2]. ROS and oxidative stress can directly damage proteins, causing them to misfold [3]. Similarly, chronic ERS activation increases the accumulation of misfolded and unfolded proteins, which then activate the unfolded protein response (UPR) through three mediators: ATF6, PERK, and IRE1 [4]. The UPR halts protein synthesis to promote correct folding and restore protein processing and secretion in order to cope with and relieve stress [5]. However, sustained activation can lead to autophagy, mitophagy, cell failure, and apoptosis [6]. Central ERS and UPR molecules include GRP78, CHOP, PERK, ATF6, IRE1, and HSP70 [7,8]. Although these molecules are predominantly intracellular, small amounts are released extracellularly and can be measured in serum [9], making them promising candidate biomarkers or therapeutic targets for conditions such as diabetes [10].

Evidence links ERS to type 2 diabetes (T2D) pathogenesis and its complications. In the liver, sustained ERS promotes hyperglycemia [11]. In the pancreas, it causes β-cell dysfunction, degeneration, and death [12]. Hyperglycemia also induces ERS in cardiomyocytes [13], endothelial cells, osteoblasts [14], podocytes, retinal cells, and neurons, contributing to acute and chronic diabetic complications [15].

Recent preclinical studies have found that ERS molecules, such as GRP78 and CHOP, are increased in cell cultures and organs of diabetic mice, such as kidney, pancreas, heart, eye, and the brain, participating in the development of organ damage induced by diabetes [16,17,18]. Additionally, the inhibition of ERS is beneficial in such organs because it reduces insulin resistance, prevents the damage and limits the progression of complications in such tissues [19,20,21,22]. Consequently, preclinical evidence indicates that the ERS pathways are key in diabetes and the organ damage induced by ERS.

However, there is no study that systematically synthesized clinical studies on the differences in serum levels of ERS markers between individuals with and without T2D, and how they correlate with markers of glucometabolic status, or on which serum ERS markers could be predictors of diabetes complications and therapeutic targets in individuals with diabetes. We performed a meta-analysis as this type of study increases the statistical power of primary studies and provides a comprehensive synthesis of the existing evidence. We hypothesized that few biomarkers of ERS are consistently increased in humans with T2D and correlate with metabolic and anthropometric variables such as fasting plasma glucose (FPG), HbA1c, Homeostasis Model Assessment (HOMA), and body mass index (BMI).

## 2. Materials and Methods

### 2.1. Protocol Registration

The protocol for this study was prospectively registered in PROSPERO (the International Prospective Register of Systematic Reviews; registration number CRD42022341343).

### 2.2. Search Strategy and Information Sources

The study investigators are specialized in Diabetes (J.R.V.B., H.G.O., L.O.), Clinical Chemistry (F.B.O., I.P.G.G.), Biomarkers (I.P.G.G.), Biology (C.C.C., A.M.H., M.I.M.R), Evidence-Based Health Care (R.J.M.P.), and Meta-analysis and Statistics (R.J.M.P., J.R.V.B.).

Before the definite literature search, we explored STRING to identify all proteins involved in the ERS and UPR and found 232 proteins involved in these processes. Then, we incorporated them into the search criteria individually or as groups of proteins, including chaperones, peroxiredoxin, and heat shock proteins [the query criteria are listed in Appendix A]. In the final meta-analysis, we included all ERS biomarkers that were measured in at least 2 studies independently of their effect size and *p*-values.

Two authors (J.R.V.B. and A.M.H.) searched the Cochrane Library, PROSPERO, Scopus, Web of Science, and PubMed without language or publication year restrictions, limiting the search to human studies. They also searched manually for relevant publications. The first search occurred on 15 June 2022, and the final updated search was conducted on 2 October 2024. If any article was in a language other than English or Spanish, we asked a native speaker to extract the data. Additionally, abstracts and unpublished studies were excluded.

This review follows the Meta-analysis of Observational Studies in Epidemiology (MOOSE) guidelines [23] and the Preferred Reporting Items for Systematic Reviews and Meta-Analyses (PRISMA) 2020 guidelines [24]. Two independent reviewers examined the abstracts while blinded to authorship, affiliations, and results (F.B.O. and A.M.H.). Full texts of eligible abstracts underwent full review. Any disagreements were resolved by two senior investigators (J.R.V.B and R.J.M.P.) who were also blinded to the study details and cause of the initial disagreement.

### 2.3. Study Selection and Outcomes

We included cross-sectional, case-control, and baseline data from cohort studies. Studies were eligible if they included patients with T2D and non-diabetic controls, and if they reported as primary outcomes the mean serum/plasma levels of ERS markers such as CHOP, PERK, ATF6, ATF4, XBP1, glucose-regulated proteins (GRP78, GRP94, and GRP170), heat shock proteins (HSP, HSP12A, HSP70, HSP72, HSP25, HSP27, HSP60, HSP60, HSP90, HSP47, HSPA8, HSP12A, and HSP65), chaperones, peroxiredoxins (PRX1, PRX2, PRX4, and PRX6), or secretagogin. Additionally, studies reporting Pearson or Spearman coefficient correlations between any ERS marker and glycemic, anthropometric, or lipid measures (secondary outcomes) were also eligible. Studies without information on the ERS biomarkers level or their correlations with glucometabolic biomarkers were excluded. We also excluded studies that lacked healthy controls, were performed in subjects with type 1 diabetes or gestational diabetes, or measured the markers of interest in a fluid or tissue other than serum/plasma.

### 2.4. Data Extraction

The following information was entered into a data extraction template (M.I.M.R. and A.M.H.) following the Cochrane Consumers and Communication Review Group’s data extraction template: author, year of publication, country where the study took place, type of study, original inclusion and exclusion criteria, the ERS marker reported, groups and the total number of participants included, sex, mean age, BMI, diabetes duration, and HbA1c levels. Correlation coefficients between ERS markers and glucometabolic, anthropometric, and lipid variables were also included. If data were missing or only presented graphically, we e-mailed the authors requesting the numerical values and, if this was not successful, we estimated the means and standard deviations from published graphs in each work using R studio v1.1.463 and the “digitize” package [25].

### 2.5. Assessment of Risk of Bias

To evaluate study quality, two reviewers (I.P.G.G. and C.C.C.) independently evaluated the bias, employing the Newcastle–Ottawa Scale for cross-sectional studies. This scale considers three dimensions: the ascertainment of exposure, the groups’ comparability, and the study groups’ selection. One star is awarded for each question in all dimensions. Studies with 6 or more stars are classified as high quality, while studies with fewer than 6 are low quality. Any disagreement in the risk of bias was resolved with the participation of a third reviewer (H.G.O.) [26].

### 2.6. Data Analysis

All statistical analyses were conducted in R studio v1.1.463 [27]. For determining differences in ERS biomarkers between T2D and healthy controls, we estimated the standardized mean difference (SMD) as the effect size measurement. The SMD was based on a random-effects model (REM) weighted by the inverse of the variance; because all studies come from different populations and random sampling is expected among them, this differs from studies performed in the same population (city, hospital). For this analysis the metacont function of “meta v4.2” package was employed. The default method for calculating confidence intervals is Hedges’ method (1981), and it is written as follows: methods.smd = ”Hedges”. Hedges (1981) calculated the exact bias in Cohen’s d, which is a ratio of gamma distributions with the degrees of freedom, i.e., total sample size minus two, as the argument. By default (argument exact.smd = FALSE), an accurate approximation of this bias provided in Hedges (1981) is utilized for Hedges’ g as well as its standard error. For Hedges’ g, the exact formulae are used to calculate the SMD as well as the standard error [28].

Additionally, the effect size was determined for correlations (Pearson or Spearman) between ERS markers and metabolic (e.g., glycemic), anthropometric (e.g., BMI), or lipid (e.g., triglycerides) variables. All effect sizes with 95% CI were estimated using the metacor function of meta package using common-effect and random-effects estimates weighting by the inverse variance for pooling. The default method for calculating effect sizes and confidence intervals was that of DerSimonian–Laird [29]. The results of pooled correlations are synthesized in a table.

We performed a sensitivity analysis to perform robustness assessment, to identify outliers, and to evaluate the model validity and the generalizability of results. This analysis works as an influence analysis in which the pooled estimates are re-calculated omitting one study at a time; this is also called “leave-one-out” analysis and helps in understanding if there is a significant influence of one study in the pooled results. A common-effects model is assumed (pooled = “common”) if a random-effects model is not assumed (pooled = “random”). For this analysis, the metainf function of the “meta v4.2” package was employed [29].

In this manuscript, we managed and analyzed the heterogeneity using the following recommendations from Cochrane Handbook for Meta-analysis: (i). performing a random-effects model; (ii). using SMD rather than other effect measures; (iii). performing a leave-one-out analysis; (iv). exploring heterogeneity by using meta-regression analysis to understand the influence of other variables on the pooled results; and (v). estimating the Egger test and applying a trim and fill analysis.

Interstudy variability was assessed using the τ2, Cochran’s Q (or χ^2^, or Chi^2^), and I^2^ statistics. For an effect size with I^2^ ≥ 40%, indicating moderate to high heterogeneity, we performed meta-regressions to explore potential sources of variability and to determine which covariates influenced ERS marker levels. This chi-squared (χ^2^, or Chi^2^) test assesses whether observed differences in results are compatible with chance alone. A low *p* value (or a large chi-squared statistic relative to its degree of freedom) provides evidence of heterogeneity of intervention effects (variation in effect estimates beyond chance). Since clinical and methodological diversity always occur in a meta-analysis, statistical heterogeneity is inevitable. Thus, the test for heterogeneity is irrelevant to the choice of analysis; heterogeneity will always exist whether we happen to be able to detect it or not using a statistical test. Methods have been developed for quantifying inconsistency across studies; these methods shift the focus from testing whether heterogeneity is present to assessing its impact on the meta-analysis. A convenient statistic for quantifying inconsistency is I^2^, which describes the percentage of variability in effect estimates that is due to heterogeneity rather than sampling error (chance) [30].

For the meta-regression, the following independent variables were examined: patient age, body mass index, and HbA1c. Meta-regressions were performed using a mixed-effects meta-regression model, Meta-regression for objects of class meta. This is a wrapper function for the R function rma in the R package metafor [31].

Publication bias was quantified through Egger’s method only for biomarkers reported in 6 or more studies, as the publication bias analysis is less reliable with fewer studies. A significant *p*-value (typically below 0.05) indicates potential publication bias, meaning the funnel plot is asymmetric and smaller studies with significant effects might be missing; essentially, the test looks for a non-zero intercept in a regression model where the effect size is regressed on its standard error (precision) across studies. We used the metabias function from the “meta” package in R statistics. The argument method.bias is “Egger”, and the test statistic is based on a weighted linear regression of the treatment effect on its standard error. The test statistic follows a t distribution with the number of studies − 2 degrees of freedom [32].

The trim and fill method was estimated to adjust for publication bias and estimate the unbiased pooled effect. We used the trimfill function from the metafor package for estimated possible “missing” studies, recalculated the pooled effect, and constructed a second funnel plot. The trimfill is a nonparametric (rank-based) data augmentation technique proposed by Duval and Tweedie (2000). The method can be used to estimate the number of studies missing from a meta-analysis due to the suppression of the most extreme results on one side of the funnel plot. This method partially examines publication bias and heterogeneity [33].

## 3. Results

### 3.1. Study Selection and Study Characteristics

A total of 1206 studies were initially found across the searched databases to identify potentially relevant studies meeting the inclusion criteria. After removing duplicates, 961 records were screened and 35 underwent full-text review for eligibility. Ultimately, 21 studies were retained for systematic review and 17 for meta-analysis (Figure 1). The reasons for excluding 13 papers are presented in Appendix A. Excluded studies either had a non-observational design or did not report serum levels of ERS markers.

The countries where the studies were conducted are presented in Table 1. In total, data from 11,953 subjects were analyzed (2224 patients with T2D and 9992 controls). The mean age at inclusion was 53.5 years and the male to female ratio was 50:50. Among diabetics, mean diabetes duration was 7.2 years, although some studies included patients with more than 10 years after diagnosis, and others had new-onset diabetes. The ERS biomarkers reported were: HSP27 (two studies), HSP70 (six studies), HSP90 (two studies), HSPA5 (two studies), GRP78 (two studies), secretagogin (two studies), and PRX4 (five studies), as well as PRX1, PRX2, and PRX6 in the same three studies (Table 1).

### 3.2. Risk of Bias

The risk of bias was assessed for all 22 studies using the Newcastle–Ottawa scale, which classified the studies as high quality (score range 6–9 out of 9 stars). The primary weakness was in the selection and definition of control groups, likely because the clinical and biochemical basal profiles of non-diabetic participants were not reported in several studies (Table 2).

### 3.3. Comparison of ERS Markers Among Diabetic and Non-Diabetic Subjects

In total, 11 markers of ERS were measured across the selected papers. Effect sizes and forest plots could be estimated for nine markers reported in at least two studies: HSPA5 (GRP78), HSP27, HSP70, HSP90, PRX1, PRX2, PRX4, PRX6, and secretagogin.

Pooled analysis using the random-effects model revealed significantly higher serum levels of HSP70 (SMD: 2.30, 95% CI: 1.13–3.46; *p* < 0.001) and secretagogin (SMD: 0.60, 95% CI: 0.19–1.01; *p* < 0.001) in T2D patients versus controls. Additionally, the pooled serum levels of PRX1 (SMD: 0.90, 95% CI: 0.55–1.25, *p* < 0.001), PRX2 (SMD: 1.21, 95% CI: 0.85–1.57, *p* < 0.001), PRX4 (SMD: 0.49, 95% CI: 0.26–0.73, *p* < 0.001), and PRX6 (SMD: 0.65, 95% CI: 0.31–1.00, *p* < 0.001) were significantly higher in patients with T2D than in controls (Figure 2). In contrast, no significant between-group differences in HSP27, HSP90, or HSPA5 were detected based on pooled estimates (Figure 3). Heterogeneity (I^2^) was low for PRX1, PRX2, PRX4, PRX6, and secretagogin, but it was high for HSP70, GRP78, HSP27, and HSP90.

### 3.4. Correlations of ERS Markers with Glucometabolic, Anthropometric, or Lipid Traits

Meta-analysis examining correlations between ERS markers and metabolic, anthropometric, and lipid traits revealed several significant associations. Secretagogin inversely correlated with HOMA-IR, yet it positively correlated with HOMA-B, HbA1c, and FPG. PRX4 negatively correlated with both HbA1c and FPG, while HSP70 positively correlated only with HbA1c. GRP78 was positively associated with FPG and BMI. No significant correlations between ERS markers and serum lipids were detected (Table 3).

### 3.5. Sensitivity Analysis

We performed a sensitivity analysis for HSP70 and PRX4 because there were enough studies (n > 3) to perform it. We found that the exclusion of each study did not affect the results direction or strength of SMD for HSP70 and PRX4 (Figure 4).

### 3.6. Meta-Regression to Explain the Heterogeneity of SMD in ERS Biomarkers

We performed a meta-regression to explain the heterogeneity between studies in HSP70 levels, as there were enough studies to support this analysis. The meta-regression revealed that BMI significantly influenced HSP70 heterogeneity (estimate: −108.3, 95% CI: −182.7 to −33.9, *p* = 0.0043), accounting for 65.28% of the variability. Specifically, lower BMI was associated with higher HSP70 levels. However, HbA1c and diabetes duration did not significantly contribute to the heterogeneity in reported HSP70 levels (Table 4). The Egger test revealed no publication bias for HSP70 (bias estimate: 2.8409, SE = 8.79; *p* = 0.760), but it was not estimated for other biomarkers because this test is not reliable when there is a limited number of studies, and we cannot discard any publication bias for the other markers.

### 3.7. Relationship of ERS Markers with Diabetes Complications

There was an insufficient number of studies to generate pooled estimates (meta-analysis) for associations between ERS markers and diabetes complications. However, this systematic review identified links between individual markers and selected complications. High serum HSP27 is independently related to subclinical atherosclerosis, as measured by increased carotid intima–media thickness [53] and to distal symmetric polyneuropathy [38]. PRX4 independently predicts cardiovascular mortality and improves the prediction of current cardiovascular risk models [52]. This molecule also predicts the development of diabetic nephropathy in cohort studies [52]. Diabetic nephropathy is also associated with elevated circulating levels of HSP27, GRP78, and CHOP [35,38]. Diabetic retinopathy is related to elevation of HSP27 and HSP70 [38,46]. 

### 3.8. Funnel Plot, Funnel Plot with Trim and Fill and Egger Test to Explain the Heterogeneity and Publication Bias in HSP70

This part of the analysis was only performed for HSP70, as these tests are not reliable when there is a limited number of studies. We constructed the funnel plot and found high visual heterogeneity, which coincided with the I^2^ metric (96%) (Figure 5, left side). We then applied a trim and fill analysis to adjust for publication bias and estimated the unbiased pooled effect for HSP70 (Figure 5, right side). However, since both graphs were very similar, the trim and fill method did not detect a significant need for adjustment for publication bias, which strongly suggested that the studies possess random variability and/or real heterogeneity. The Egger test revealed no publication bias for HSP70 (bias estimate: 2.8409, SE = 8.79; *p* = 0.760). 

## 4. Discussion

Our findings demonstrate that of the 232 proteins involved in ERS and unfolded protein response [56], only 11 ERS biomarkers have been measured in the serum of patients with T2D, and 6 (HSP70, GRP78, PRX1, PRX2, PRX4, PRX6, and SCGN) were consistently and significantly elevated compared with controls. These biomarkers positively correlated with BMI, glycated hemoglobin, fasting plasma glucose, and HOMA-IR. These findings underscore the importance of these seven ERS proteins as potential biomarkers clinically relevant in T2D, since they differ significantly between T2D and controls and correlate with key adiposity and glycemic control parameters.

Based on the SMD, the biomarkers with the strongest associations with T2D were HSP70, PRX2, and PRX1 (SMD > 0.8), and those with moderate elevations were PRX6, secretagogin, and PRX4 (SMD > 0.4). Although only two or three studies were performed for several biomarkers [PRX1, PRX2, PRX4, PRX6, and secretagogin], the results might be reliable, as their heterogeneity is lower than 33% and their SMD indicates a moderate to strong difference between T2D and controls. In the case of HSP70, the meta-regression allowed us to explain part of the heterogeneity by differences in BMI between subjects. Furthermore, although it is suggested that an I^2^ may be a drawback or affect the validity of the meta-analysis findings, the validity of each study individually is not affected [57]. As can be seen in Figure 2A, five out of six studies reported significant differences in HSP70 levels (higher) in patients with T2D than in controls. Moreover, I^2^ is not an absolute measure of heterogeneity, as Borenstein, Higgins, and Hedges have demonstrated [58].

The observed increases in these specific ERS markers likely reflect compensatory cellular responses to mounting stress induced by hyperglycemia [59]. For example, the serum elevation of HSP70 in subjects with diabetes may indicate its increased production to handle accumulating stress, since this protein helps to refold or degrade misfolded proteins [60]. The clinical relevance of heat shock proteins in diabetes is further evidenced by studies showing higher levels of HSP70, HSP27, HSPA5, and HSPA8 in target organs affected by diabetic complications, as identified through this systematic review and meta-analysis [35,38,46,53,54,61]. Previous studies demonstrated that abnormal expression levels of HSP70 induce inflammation and affect insulin sensitivity as well as glycemic control, making it a potential target for diabetes management [62]. Further evidence links HSP70 to the development of diabetes and vascular diabetic complications through the Toll-like receptor 4 (TLR4) pathway [63]. Additionally, primary studies included in this meta-analysis [44,45,46,47,48,49] and our meta-analysis found a consistent and significant correlation of HSP70 with triglycerides and HbA1c, highlighting its clinical relevance.

PRX1 and PRX4 are antioxidant enzymes that counteract oxidative stress and are highly expressed in tissues affected by diabetes, such as pancreas, liver, and heart. These enzymes decrease the synthesis of mediators of ERS, such as XBP-1 and CHOP [64]. Meanwhile, PRX4 and PRX6 provide additional cytoprotection by detoxifying peroxides and modulating signaling pathways, such as the NF-κB pathway, and phospholipid metabolism [65]. Furthermore, PRX4 is located in the ER compartment, where it reduces oxidized proteins, induces disulfide bond formation in proteins, and helps in antioxidant defense [66]. Consequently, the observed elevated levels of circulating peroxiredoxins among patients with T2D may represent a compensatory adaptation that counteracts elevated systemic oxidative stress in this population and maintains cellular homeostasis [67]. Further characterization of the impact of heightened PRX1, PRX4, and PRX6 activity across tissues may provide insights into the role of these enzymes in disease progression and complications of diabetes (Figure 5). PRX4 is also elevated in polycystic ovary syndrome, and it is negatively correlated with total oxidant status but positively correlated with insulin and HOMA-IR [68]. Additionally, PRX4 is independently associated with cardiovascular mortality in patients with T2D, making it a new non-traditional risk factor that modestly improves the risk prediction [52]. Furthermore, it predicts new-onset diabetic nephropathy [55]. In conjunction with our findings, it supports the hypothesis that PRX4 is a prospective biomarker of diabetes complications.

Secretagogin (SCGN) is a protein that is synthesized at high levels by pancreatic β-cells and neuroendocrine cells and that has recently emerged as a specific marker of ERS in β-cells [39]. Beyond modulating insulin secretion via Ca^2+^-dependent pathways and promoting insulin vesicle transport [69], SCGN also facilitates protein folding, enhances β-cell survival and proliferation, and protects key proteins such as Pdx-1 from proteasomal degradation, thereby supporting the maintenance of β-cell function [70]. Consistent with these functions, the findings of this study demonstrate parallel increases in circulating SCGN levels with worsening glycemic control and insulin resistance, as evidenced by positive correlations with HOMA-IR, HbA1c, and fasting plasma glucose. This observation is supported by recent in vitro observations where SCGN was noticeably elevated in response to endoplasmic reticulum stressors and cytokines but not in response to glucose-induced insulin secretion in β-cells [39]. Meanwhile, negative correlations between SCGN and HOMA-B suggest that as SCGN rises, insulin secretion decreases (Figure 5). Future studies should confirm if SCGN is an ERS biomarker specific to pancreatic β-cells.

All significant correlations of ERS biomarkers with BMI, HbA1c, glucose, HOMA-B, and HOMA-IR point toward the clinical relevance of HSP70, GRP78, PRX1, PRX2, PRX4, PRX6, and SCGN, as they are not only elevated in patients with diabetes but also correlate with glucometabolic variables (Figure 5). Remarkably, secretagogin and peroxiredoxins correlated with glycemic parameters but not with serum lipids. On the other hand, GRP78 correlated with BMI and serum lipids in addition to FPG. Such findings indicate that some ERS biomarkers might be linked to glycemic and anthropometabolic abnormalities. Preclinical studies strongly support the role of GRP78 in the pathogenesis of obesity and T2D, suggesting it is a candidate for treating metabolic diseases [71].

It is essential to highlight that although no meta-analysis was performed to evaluate the association of ERS biomarkers with diabetes complications due to the paucity of studies, the evidence from single studies suggests an independent association of serum PRX4 levels with mortality in T2D [52] and incident diabetes [41]. Another study observed that GRP78 is altered in diabetic kidney disease [35], while HSP27 has been linked to diabetic nephropathy [38], distal symmetrical polyneuropathy [72] and subclinical atherosclerosis [53]. Thus, serum ERS biomarkers are not only dysregulated in diabetes, but they are also related to chronic complications of diabetes and mortality. Recent studies provide evidence that circulating GRP78 and CHOP levels are significantly higher in heart failure (HF) patients, that they are related to disease severity, and that they possess moderate predictive values for HF [73]. Furthermore, GRP78 is implicated in the development of diabetic kidney disease (DKD) and peritubular fibrosis in vivo since the administration of canagliflozin reduces ER stress impartment, protects proximal tubular cells, and restores subcellular localization of GRP78, SGLT2, and integrin-β1, reducing the fibrosis in DKD [74]. Additionally, the PERK/eIF2α/ATF4/CHOP branch of ERS promotes vascular calcification [75], and ERS is linked to vascular insulin resistance affecting vascular relaxation through the activation of GRP78, and CHOP, p-JNK/JNK, and the IRS-1PI3K/Akt/eNOS pathway [22].

We should stress that the serum ERS molecules found here are candidates for further clinical validation since they fulfill several hallmarks of ideal biomarkers, including their capability to discriminate diseased from healthy subjects, the fact that they are easily obtained from an accessible tissue (blood), and their reproducible measurement. Additionally, they are related to glucometabolic variables and they have biological plausibility (Figure 6).

At present, there is a lack of data regarding changes in ERS induced by diabetes treatment, and there are not enough longitudinal studies evaluating the utility of ERS markers in identifying the presence of diabetes or its complications. Thus, it is desirable to evaluate whether serum ERS biomarkers change in response to treatment or might be reflective of diabetes complications. Furthermore, it is essential to explore whether therapeutic modulation of ERS impacts glycemic control and prevents diabetes complications. It is important to recognize that ERS serum proteins do not replace traditional clinical biomarkers such as HbA1c, HOMA-IR, and HOMA-B. However, they may provide additional knowledge on the underlying pathophysiological mechanisms of T2D. Additionally, it remains necessary to evaluate whether ERS proteins are surrogates of treatment response at a cellular level or if they reflect complications in specific organs, in case ERS biomarkers specific to some organs and tissues are discovered. In any case, further studies should be performed across different subpopulations and disease stages.

Some preclinical studies have tested the effect of ERS inhibitors [morin, tauro-ursodeoxycholic acid, quercetin, and 4-phenyl butyric acid] on diabetes, demonstrating positive glycemia reduction through an insulin-mimetic effect, the inhibition of gluconeogenesis, and the promotion of β-cell survival, insulin secretion, and insulin sensitivity [76,77,78]. Hypoglycemic agents currently used in clinical practice [such as pioglitazone, exenatide, liraglutide, and dapagliflozin] inhibit components of ERS in vitro and in vivo, and liraglutide reduces endothelial ERS in patients with diabetes [19,79,80,81]. However, there are no evaluations of the impact of the latter drugs on the serum levels of ERS proteins in patients with T2D, and it is unclear if the long-term benefits of such drugs are due, at least in part, to the inhibition of ERS.

Other recent preclinical studies have revealed that molecules participating in ERS are promising therapeutic targets. Nevibolol inhibits ERS, alleviating endothelial insulin resistance [22]. Linagliptin decreased the Chop mRNA level in diabetic mouse hearts [19]. Vitamin D alleviates ERS in mature β-cells [82]. Additionally, in vitro and in vivo studies demonstrated the ability of Asian and Middle Eastern traditional herbal preparations and individual compounds extracted from some foods to decrease the expression of Ire1, Atf6, Atf4, Grp78, caspase-12, and Chop in tissues and cell cultures from diabetic rats, improving glucose metabolism and insulin sensitivity [83,84,85]. Consequently, the modulation of ERS pathways holds promise for preventing or treating diabetes and its complications.

We recognize some limitations of systematic reviews and meta-analyses, including their reliance on secondary data, the potential for publication bias, and biases in the analysis and reporting of primary studies, as well as the heterogeneity across the included studies. Furthermore, the small sample size of the primary studies and their heterogeneity might affect the generalizability of the findings. However, we addressed the potential limitations of this study in several ways. Firstly, we assessed the quality assessment of all included studies and found that they were all of good quality according to the Newcastle–Ottawa classification. The sensitivity analysis demonstrated the consistency of the effect size, even when studies were excluded one by one, reinforcing the validity of the results, the robustness of the findings, and the generalizability of the conclusions. Additionally, we did not detect significant publication bias for HSP70 employing the Egger test (Appendix A). In addition, the heterogeneity observed in HSP70 serum levels was mainly explained (65.28%) by differences in participant BMI via meta-regression analysis. Since the funnel plot detected important visual heterogeneity in HSP70, we performed a trim and fill analysis, which confirmed a real heterogeneity and random variability in the levels of HSP70, but not publication bias. Consequently, further actions should be taken to standardize its measurement [as has been achieved for HbA1c] before it can be considered a marker for clinical use.

### Future Research Directions

Future studies should be performed to prospectively evaluate ERS biomarkers in larger and more diverse populations to establish their utility in monitoring disease, evaluating treatment response, and detecting complications. More studies will help us to determine if there is a difference in the performance of ERS biomarkers in populations with a high prevalence of diabetes, such as MENA, Asian, and Latin-American populations. Further studies are needed to assess the mechanistic roles of each ERS component in the development of T2D complications. Additionally, new research should be implemented to evaluate the efficacy of other drugs, plant compounds, and food components that modulate ERS components in improving glycemic control, reducing the need for hypoglycemic therapy, and preventing diabetes complications. Furthermore, it is necessary to standardize the measurement of ERS biomarkers [as has been achieved for HbA1c] before they can be considered markers for clinical use.

## 5. Conclusions

Six ERS biomarkers (HSP70; peroxiredoxin-1, -2, -4, -6; and SCGN) were consistently elevated in patients with T2D and showed a significant correlation with BMI, HbA1c, FPG, HOMA-IR, and HOMA-B. Thus, they are candidates for further clinical validation as biomarkers. Future studies in larger and more diverse populations should prospectively evaluate their utility in monitoring disease, evaluating treatment response, and predicting complications. Additionally, further studies are needed to assess the mechanistic roles of ERS components in T2D complications and the impact of their therapeutic modulation. We acknowledge the study’s limitations, including the primary studies’ heterogeneity and limited sample sizes. More articles must be published to enable more robust conclusions and allow for the implementation of other meta-analytic approaches, such as subgroup analysis.

## Figures and Tables

**Figure 1 antioxidants-13-01564-f001:**
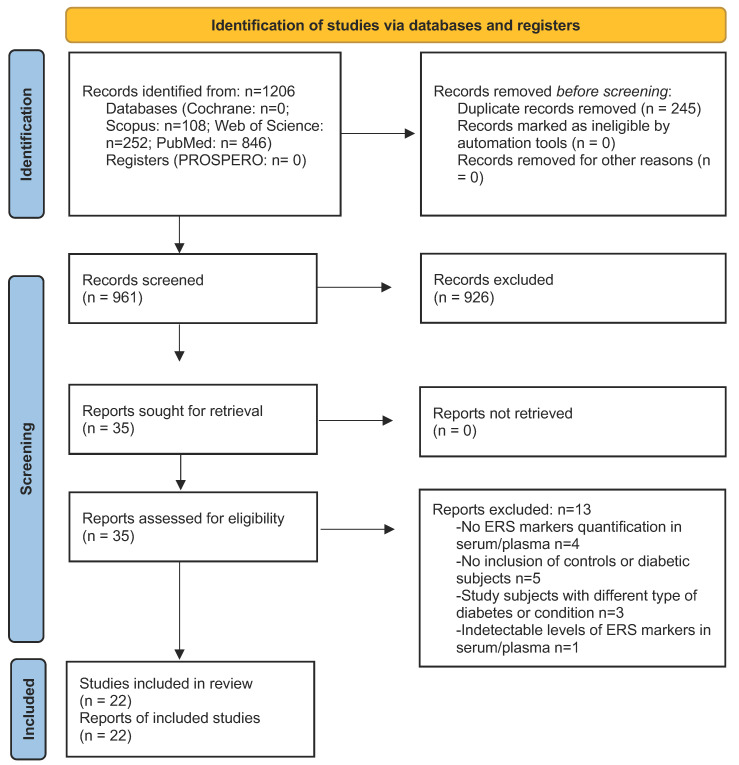
PRISMA flow diagram of the study.

**Figure 2 antioxidants-13-01564-f002:**
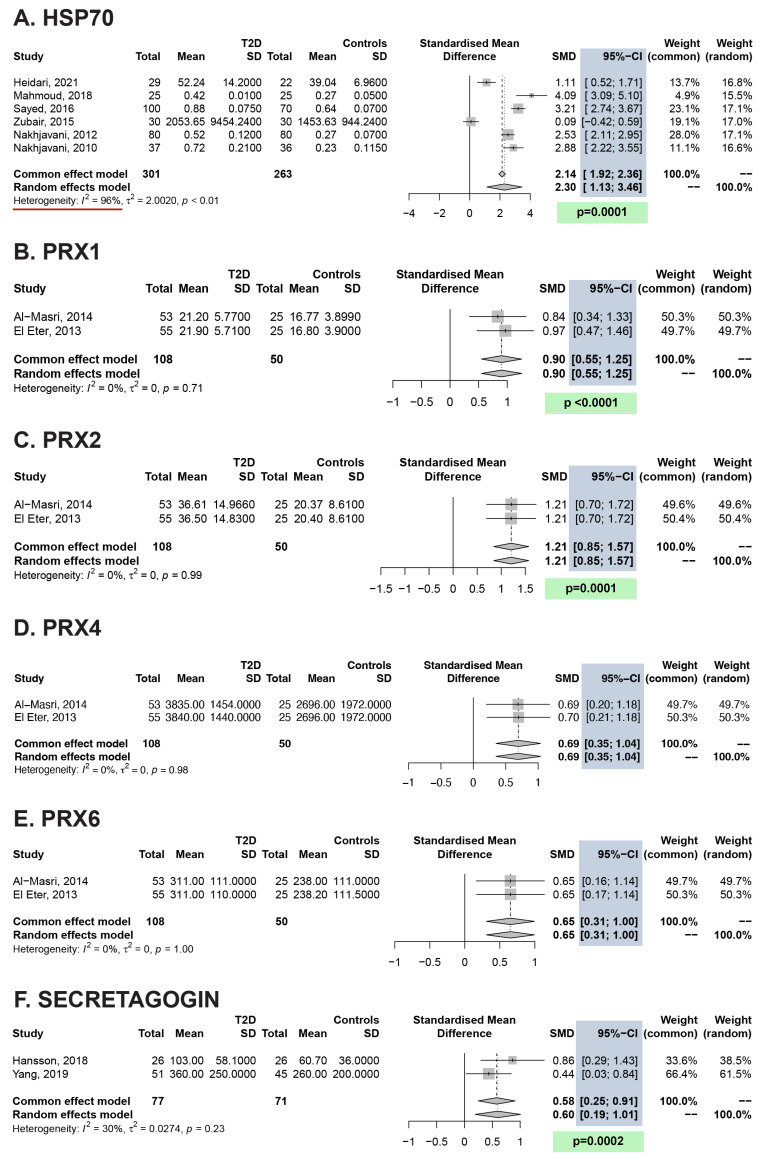
ERS biomarkers significantly increased in T2D. SD: standard deviation; SMD: standardized mean difference; CI: confidence interval. We highlighted in blue the 95% confidence intervals, in green the significant *p*-values, and in red those studies with I^2^ [heterogeneity] greater than 40% [39,40,42,43,44,45,46,47,48,49].

**Figure 3 antioxidants-13-01564-f003:**
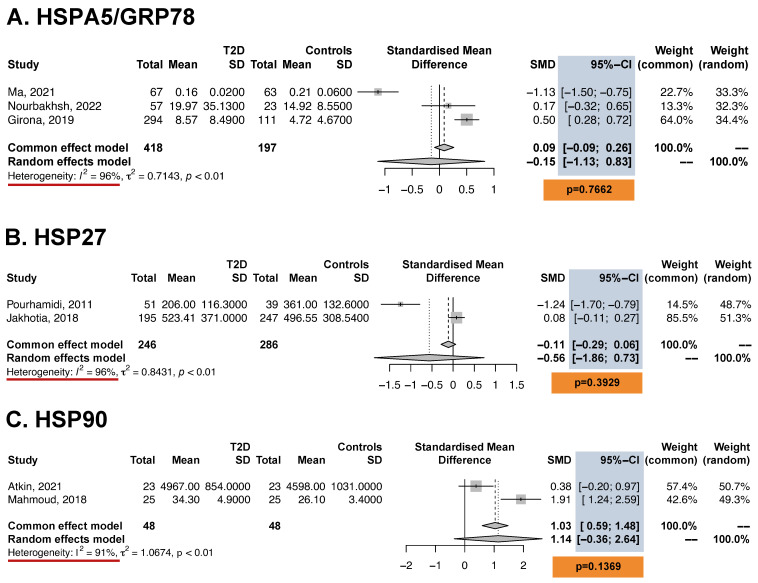
ERS biomarkers not significantly associated with T2D. SD: standard deviation; SMD: standardized mean difference; CI: confidence interval. We highlighted in blue the 95% confidence intervals, in orange the non-significant *p*-values, and in red those studies with I^2^ [heterogeneity] greater than 40% [34,35,36,37,38,45,50].

**Figure 4 antioxidants-13-01564-f004:**
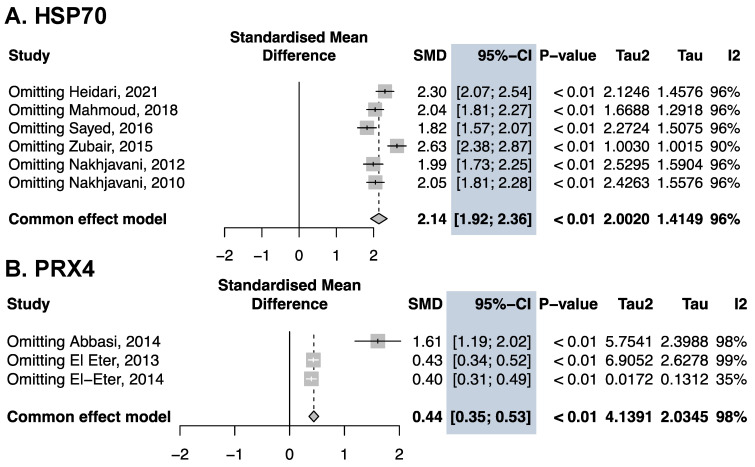
Forest plot of the sensitivity analysis for HSP70 and PRX4 shows the effect size’s consistency and discards outliers. SD: standard deviation; SMD: standardized mean difference; CI: confidence interval [blue] [44,45,46,47,48,49].

**Figure 5 antioxidants-13-01564-f005:**
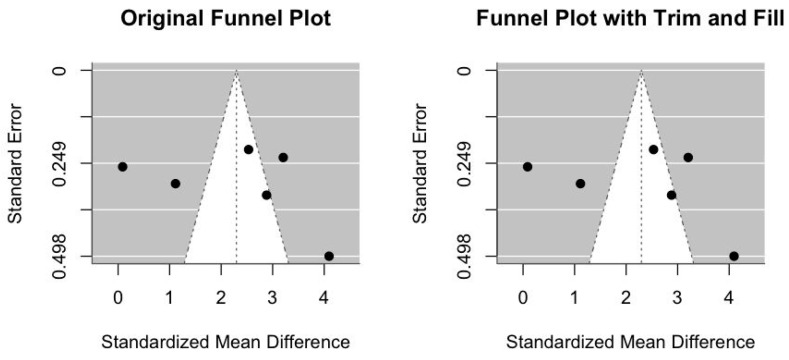
The funnel plot and funnel plot with trim and fill for evaluating visual heterogeneity for HSP70.

**Figure 6 antioxidants-13-01564-f006:**
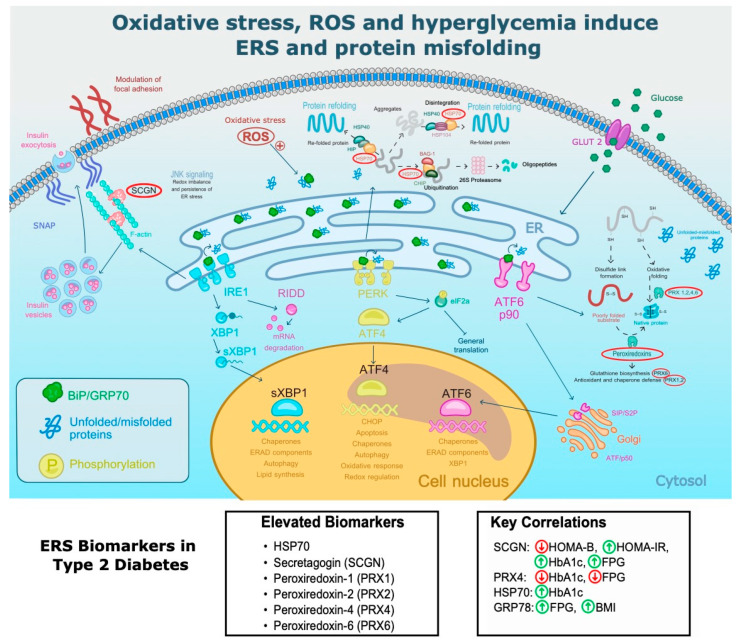
Components of the ERS pathway where HSP70, PRX and secretagogin participate. Stimuli like nutrient deprivation, oxidative stress, reactive oxygen species (ROS), insulin resistance, and hyperglycemia induce protein misfolding. Consequently, the machinery of the ERS is activated to counteract stress. Molecules such as peroxiredoxins, HSP, and secretagogin help to cope with ER stress and to refold misfolded or unfolded proteins. Peroxiredoxins help to refold proteins and contribute to glutathione biosynthesis, as well as antioxidant and chaperone defense. HSP-70 contributes to protein refolding, disintegration of protein aggregates, and the ubiquitination of misfolded proteins to degrade proteins in the proteasome. Meanwhile, SCGN facilitates protein folding, promotes insulin release, enhances β-cell survival and proliferation, and protects critical proteins like Pdx-1 from proteasomal degradation. ER: endoplasmic reticulum. Red circles indicate important molecules involved in ERS in T2D.

**Table 1 antioxidants-13-01564-t001:** General characteristics of the included studies.

Author, Year	Country	n T2D/Ctrls/Total	Mean Age	Mean BMI	Male: Female	Evolution of T2D, y	Biomarkers Measured	Outcome/Phenotype
Atkin, 2021 [34]	UK	23/23/46	62.0	30.0	48/52	4.5	HSP90	T2D
Ma, 2021 [35]	China	67/63/130	57.0	24.5	55/45	12.7	HSPA5, CHOP, GRP78	T2D, DNe
Nourbakhsh, 2022 [36]	Iran	57/23/80	51.8	28.7	37/63	-	HSPA5, GRP78, TRB3	T2D
Pourhamidi, 2011 [37]	Sweden	51/39/90	61.0	27.7	54/46	7.2	HSP27	T2D
Jakhotia, 2018 [38]	India	195/247/442	55.7	26.4	48/52	7.2	HSP27	T2D, DRe, DNe, DNu
Hansson, 2018 [39]	US	26/26/52	47.5	29.8	75/25	-	SCGN	T2D
Yang, 2019 [40]	China	51/54/105	53.2	24.8	42/58	-	SCGN	T2D
Abbasi, 2014 [41]	The Netherlands	496/7477/7973	48.9	26.0	49/51	0 *	PRX-4	T2D
Al-Masri, 2014 [42]	Saudi Arabia	53/25/78	57.6	-	47/53	8.5	PRX-1,2,4,6	T2D
El Eter, 2015 [43]	Saudi Arabia	53/25/78	57.7	26.8	58/42	8.5	PRX-1,2,4,6	T2D
Heidari, 2019 [44]	Iran	29/22/51	52.7	28.7	0/100	6.6	HSP70	T2D
Mahmoud, 2018 [45]	Kuwait	25/25/50	48.7	27.6	61/39	6.1	HSP70, HSP90	T2D
Sayed, 2016 [46]	Egypt	100/70/170	53.5	27.1	67/33	11	HSP70	T2D, DRe
Zubair, 2015 [47]	India	60/30/90	49.1	22.5	59/41	≤10	HSP70	T2D
Nakhjavani, 2012 [48]	Iran	80/80/160	49.3	27.5	50/50	-	HSP70	T2D
Nakhjavani, 2010 [49]	Iran	37/36/73	49.0	28.3	50/50	>5	HSP70	T2D
Girona, 2019 [50]	Spain	295/9/304	60.0	27.5	49/51	-	HSPA5	T2D
Yuan, 2011 [51]	New Zealand	40/41/81	53.0	-	55/45	<1	HSP60	T2D
Gerrits, 2014 [52]	The Netherlands	137/1024/1161	67.0	28.4	45/55	4.0	PRX-4	CV mortality
Wang, 2020 [53]	China	76/110/186	67.0	24.5	40/60	8.2	HSP27, GRP78	sAtherosc
Moin, 2021 [54]	Qatar	10/7/17	46.5	32.5	65/35	3.3	HSPA8	Hypoglyc
Bourgonje, 2024 [55]	The Netherlands	257/279/536	64	29.5	48/52	3.0	PRX4	DNe

-: not reported; Ctrls: controls; y: years; DRe: diabetic retinopathy; DNe: diabetic nephropathy; DNu: diabetic neuropathy; GRP78 is also named BiP or HSPA5; CV: cardiovascular; sAtherosc: subclinical atherosclerosis; Hypoglyc: hypoglycemia. * Newly diagnosed diabetes.

**Table 2 antioxidants-13-01564-t002:** Newcastle–Ottawa scale for risk of bias evaluation.

Study		Selection				Comparability		Exposure			
**Author**	Year	Is the Case Definition Adequate?	Representativeness of the Cases	Selection of Controls	Definition of Controls	Comparability of C&C on the Basis of the Design or Analysis		Ascertainment of Exposure	Same Method of Ascertainment for C&C	Non-Response Rate	Stars
Atkin [34]	2021	*	*	*	*	*	-	*	*	*	8
Ma [35]	2021	*	*	*	*	*	*	*	*	*	9
Nourbakhsh [36]	2022	*	-	-	-	*	*	*	*	*	6
Pourhamidi [37]	2011	*	*	*	*	*	*	*	*	*	9
Jakhotia [38]	2018	*	*	-	-	*	-	*	*	*	6
Hansson [39]	2018	*	*	*	*	*	*	*	*	*	9
Yang [40]	2019	*	-	-	*	*	*	*	*	*	7
Abbasi [41]	2014	*	*	*	*	*	*	*	*	*	9
Al-Masri [42]	2014	*	*	*	-	-	*	*	*	*	7
El Eter [43]	2015	*	*	*	*	-	*	*	*	*	8
Heidari [44]	2019	*	*	-	*	*	*	*	*	*	8
Mahmoud [45]	2018	*	*	-	*	*	*	*	*	*	8
Sayed [46]	2016	*	*	-	-	*	-	*	*	*	6
Zubair [47]	2015	*	*	*	-	*	*	*	*	*	8
Nakhjavani [48]	2012	*	*	*	*	*	*	*	*	*	9
Nakhjavani [49]	2010	*	*	*	*	*	*	*	*	*	9
Girona [50]	2019	*	*	*	*	*	*	*	*	*	9
Yuan [51]	2011	*	*	*	*	*	*	*	*	*	9
Gerrits [52]	2014	*	*	*	*	*	*	*	*	*	9
Wang [53]	2020	*	*	*	*	*	*	*	*	*	9
Moin [54]	2021	*	-	*	*	*	*	*	*	*	8
Bourgonje [55]	2024	*	*	*	*	*	*	*	*	*	9

C&C: cases and controls. * A star is awarded for each criterion that the study meets. “-“ A middle dash indicates that the study did not fulfill the that criterion.

**Table 3 antioxidants-13-01564-t003:** Pooled significant correlations of ERS markers with anthropometabolic (AM) variables.

AM Parameters Related to ERS	ERS Marker-AM Parameter	n	Pooled R, 95% CI by CEM	Pooled R, 95% CI by REM	I^2^
Only glycemic parameters	Secretagogin-HOMA-B	178	−0.23 (−0.37 to −0.09) *	−0.27 (−0.50 to −0.1) *	63%
	Secretagogin-HbA1c	178	0.28 (0.13 to 0.41) *	0.32 (0.04 to 0.55) *	68%
	Secretagogin-FPG	178	0.32 (0.18 to. 0.45) *	0.34 (0.15 to 0.50) *	37%
	Secretagogin-HOMA-IR	178	0.19 (0.04 to 0.33) *	0.19 (0.04 to 0.33) *	0%
	PRX4-HbA1c	166	−0.21 (−0.36 to −0.06) *	0.21 (−0.40 to 0.00) *	47%
	PRX4-FPG	166	−0.22 (−0.36 to −0.06) *	−0.22 (−0.36 to −0.06) *	0%
	PRX2-HOMA-IR	168	0.15 (0.00 to 0.30) *	0.15 (−0.09 to 0.38)	61%
Glycemic and lipidic parameters	GRP78-FPG	485	0.28 (0.20 to 0.36) *	0.28 (0.20 to 0.36) *	0%
	GRP78-BMI	485	0.28 (0.20 to 0.36) *	0.23 (0.03 to 0.41) *	67%
	GRP78-LDLc	485	0.28 (0.20 to 0.36) *	0.13 (−0.34 to 0.55)	94%
	GRP78-Total cholesterol	485	0.34 (0.26 to 0.42) *	0.24 (−0.12 to 0.54)	89%
	GRP78-Triglycerides	485	0.34 (0.26 to 0.42) *	0.24 (−0.12 to 0.54)	89%
	HSP70-Triglycerides	260	0.35 (0.24. to 0.45) *	0.29 (−0.18 to 0.65)	93%
	HSP70-HbA1c	260	0.42 (0.32 to 0.52) *	0.39 (0.13 to 0.60) *	79%

AM: anthropometabolic; CEM: common-effects model; REM: random-effects model; CI: confidence interval; HOMA-B: Homeostasis Model Assessment of β-cell Function; HOMA-IR: Homeostasis Model Assessment of Insulin Resistance; FPG: fasting plasma glucose; BMI: body mass index; LDLc: low-density lipoprotein cholesterol; PRX: peroxiredoxin; GRP78: glucose-regulated protein 78; * Statistically significant.

**Table 4 antioxidants-13-01564-t004:** Results of the meta-regression explaining the variability in HSP70.

	Estimate	SE	Z-Value	*p*-Value	CI Lower Bound	CI Upper Bound
Intercept	2845.518	1174.459	2.422	0.0154	543.620	5147.416
BMI	−108.282	37.966	−2.852	0.004	−182.694	−33.870
HbA1c	45.616	215.117	0.212	0.832	−376.006	467.238
Evolution of diabetes	−36.593	76.036	−0.481	0.630	−185.622	112.435

SE: standard error; CI: confidence interval.

## Data Availability

Data will be made available on request.

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
