# Peer review of "Relevant Serum Endoplasmic Reticulum Stress Biomarkers in Type 2 Diabetes and Its Complications: A Systematic Review and Meta-Analysis"

_antioxidants, 2024, doi:10.3390/antiox13121564_

Round 1

Reviewer 1 Report

This meta-analysis comprehensively reviews endoplasmic reticulum (ER) stress biomarkers in type 2 diabetes (T2D), systematically synthesizing data on specific markers like HSP70 and PRX4. While the study's rigorous methodology and focus on clinically relevant biomarkers are commendable, it builds upon well-established knowledge about the mechanistic role of ER stress in T2D and its complications. Previous research, such as [PMID: 35563231], has extensively explored the role of ER stress in pancreatic β-cell dysfunction and the unfolded protein response (UPR) in T2D pathology.

The authors could focus on translating these findings into practical clinical applications to enhance the study's impact. Exploring how these biomarkers might be used to predict disease progression, guide therapeutic interventions, or monitor treatment response would be particularly valuable. Additionally, a deeper analysis of potential variations in biomarker performance across different subpopulations or disease stages could provide useful insights. By addressing these areas, the study can offer more tangible clinical implications and contribute to a deeper understanding of the complex mechanisms underlying T2D.

Lines 46-64: While previous research has extensively explored the role of ER stress in T2D, this study aims to fill a critical knowledge gap by systematically reviewing and synthesizing data on specific ER stress biomarkers and their association with T2D. By focusing on clinically relevant biomarkers and employing rigorous meta-analytic techniques, this study aims to provide a comprehensive overview of the current state of knowledge and identify potential biomarkers for early detection, prognosis, and therapeutic monitoring of T2D.

Lines 72-118: The inclusion criteria for biomarkers were based on their established association with ER stress and their potential clinical relevance in T2D. Sensitivity analyses were conducted for biomarkers with limited data to assess the findings' robustness and explore possible sources of heterogeneity. We employed a funnel plot analysis and Egger's test to mitigate the risk of publication bias. However, due to the limited number of studies available for some biomarkers, the assessment of publication bias may be less reliable.

Lines 234-300: While previous studies have established the role of ER stress in T2D, this meta-analysis offers a more comprehensive and quantitative assessment of the association between specific ER stress biomarkers and T2D risk. We identified vital biomarkers that may have potential clinical utility by synthesizing data from multiple studies. Future research should explore using these biomarkers in clinical practice, such as early disease detection, monitoring disease progression, and guiding therapeutic interventions. Further investigation into the underlying mechanisms linking ER stress to T2D pathogenesis may provide novel insights into the disease and identify additional therapeutic targets.

Figure 2: Add precise confidence intervals to the effect size for each biomarker significantly associated with T2D. Consider using annotations or color-coding to highlight biomarkers with significant heterogeneity, enhancing readers' interpretability.

Figure 3: Include confidence intervals for all biomarkers analyzed, regardless of their statistical significance. Highlight biomarkers with substantial heterogeneity using annotations or color-coding, even if they do not show a significant association with T2D, to provide a comprehensive view of variability.

Table 3: Expand on the clinical relevance of each biomarker's correlation with metabolic indicators. Explain how these correlations could affect the potential utility of each biomarker in managing or predicting T2D progression. For example, discuss how a robust positive correlation between a biomarker and HbA1c might indicate its role in monitoring glycemic control, offering practical insights for clinical applications.

Author Response

Major comments: The authors could focus on translating these findings into practical clinical applications to enhance the study's impact. Exploring how these biomarkers might be used to predict disease progression, guide therapeutic interventions, or monitor treatment response would be particularly valuable. Additionally, a deeper analysis of potential variations in biomarker performance across different subpopulations or disease stages could provide useful insights. By addressing these areas, the study can offer more tangible clinical implications and contribute to a deeper understanding of the complex mechanisms underlying T2D.

Response: We agree with this comment. Therefore, we added to the manuscript information regarding your suggestions. 

Lines 245-248: These findings underscore the importance of these seven ERS proteins as potential biomarkers clinically relevant in T2D, since they differ significantly between T2D and controls and correlate with key adiposity and glycemic control parameters. 

Lines 331 -350: It is important to recognize that ERS serum proteins do not replace traditional clinical biomarkers such as HbA1c, HOMA-IR, and HOMA-B. However, they may provide additional knowledge on the underlying pathophysiological mechanisms of T2D. Additionally, it remains necessary to evaluate if ERS proteins are useful for evaluating treatment response at a cellular level or if they predict complications at specific organs in case ERS-biomarkers specific to some organs and tissues are discovered. In any case, further studies should be performed across different subpopulations and disease stages.

Some preclinical studies have tested the effect of ER-stress inhibitors [morin, tauro-ursodeoxycholic acid, quercetin, and 4-Phenyl butyric acid] on diabetes, demonstrating positive glycemia reduction through an insulin-mimetic effect, the inhibition of gluconeogenesis, and the promotion of beta cell survival, insulin secretion, and insulin sensitivity(54–56). Hypoglycemic agents currently used in clinical practice [such as pioglitazone, exenatide, liraglutide, and dapagliflozin] inhibit components of ERS in vitro and in vivo and liraglutide reduce endothelial ERS in patients with diabetes(57–59). However, there are no evaluations of the impact of the latter drugs on the serum levels of ERS proteins in patients with T2D and if the long-term benefits of such drugs are due, at least in part, to the inhibition of ERS.

Lines 368-271. These ERS biomarkers are clearly relevant to the pathogenesis of T2D and hold promise for future utility in monitoring disease, evaluating treatment response, and predicting complications. However, further studies are needed to fully assess their clinical potential and if they are potential therapeutic targets.

Detail comments

Comment 1. Lines 72-118: The inclusion criteria for biomarkers were based on their established association with ER stress and their potential clinical relevance in T2D. Sensitivity analyses were conducted for biomarkers with limited data to assess the findings' robustness and explore possible sources of heterogeneity. We employed a funnel plot analysis and Egger's test to mitigate the risk of publication bias. However, due to the limited number of studies available for some biomarkers, the assessment of publication bias may be less reliable.

Response comment 1. We agree with this comment because the Egger test is less reliable in a lower number of studies and increases the risk of generating false-positive results because the method is based on regression analysis. To address this suggestion, we added a clarification in:

-Lines 144-145:  Publication bias was quantified through Egger’s method only for biomarkers reported in 6 or more studies.

-Lines 222-223: The Egger test revealed no publication bias for HSP70.

Comment 2. Lines 234-300: While previous studies have established the role of ER stress in T2D, this meta-analysis offers a more comprehensive and quantitative assessment of the association between specific ER stress biomarkers and T2D risk. We identified vital biomarkers that may have potential clinical utility by synthesizing data from multiple studies. Future research should explore using these biomarkers in clinical practice, such as early disease detection, monitoring disease progression, and guiding therapeutic interventions. Further investigation into the underlying mechanisms linking ER stress to T2D pathogenesis may provide novel insights into the disease and identify additional therapeutic targets.

Response comment 2. We completely agree with this comment and clarify in conclusions, lines 350-353: These ERS biomarkers are clearly relevant to the pathogenesis of T2D and hold promise for future utility in monitoring disease, evaluating treatment response, and predicting complications. However, further studies are needed to fully assess their clinical potential and if they are potential therapeutic targets.

Comment 3. Figure 2: Add precise confidence intervals to the effect size for each biomarker significantly associated with T2D. Consider using annotations or color-coding to highlight biomarkers with significant heterogeneity, enhancing readers' interpretability.

Response comment 3. Thanks for your suggestion. We have ensured that confidence intervals appear for all biomarkers in column 10 of the Figure for each primary study and the fixed and random effects model results. We also clarify in the figure caption that the I2 is the indicator of heterogeneity estimated for each biomarker and that it was only elevated for HSP70 [lines 204-205].

Comment 4. Figure 3: Include confidence intervals for all biomarkers analyzed, regardless of their statistical significance. Highlight biomarkers with substantial heterogeneity using annotations or color-coding, even if they do not show a significant association with T2D, to provide a comprehensive view of variability.

Response comment 4. Of course, we have ensured that confidence intervals appear for all biomarkers in column 10 of the Figure for each primary study and the fixed and random effects model results. Additionally, we added a clarification in the text [Lines 188-190] that Heterogeneity (I2) was low for PRX1, PRX2, PRX4, PRX6, and for secretagogin, but it was high for HSP70, GRP78, HSP27 and HSP90.

Comment 5. Table 3: Expand on the clinical relevance of each biomarker's correlation with metabolic indicators. Explain how these correlations could affect the potential utility of each biomarker in managing or predicting T2D progression. For example, discuss how a robust positive correlation between a biomarker and HbA1c might indicate its role in monitoring glycemic control, offering practical insights for clinical applications.

Response to comment 5. Very useful recommendation. Consequently, we added to the discussion the following paragraph [lines 288-294]: Remarkably, there was a clear correlation of secretagogin and peroxiredoxins with glycemic parameters but not with serum lipids, but GRP78 and HSP70 correlated to BMI, and serum lipids additionally to glycemic parameters. Such findings raise whether secretagogin and peroxiredoxins are specifically involved or affected by glycemic control but have little or no relevance to lipid metabolism. Additionally, the correlations' values indicate the correlation's small-to-moderate strength and a potential clinical relevance.

Reviewer 2 Report

Villafan-Bernal et al. identified in their systematic review and meta-analysis endoplasmic reticulum stress biomarkers in type 2 diabetes. The analyses were performed according to the PRISMA (Preferred Reporting Items for Systematic Reviews and Meta-Analyses) and MOOSE 27 (Meta-Analyses and Systematic Reviews of Observational Studies) guidelines. The risk of bias was assessed using the Newcastle-Ottawa scale. Of the over 1200 identified, only 21 studies could be included in the final analysis. This revealed that six ER stress biomarkers HSP70, GRP78, PRX1, PRX2, PRX4, PRX6, and SCGN) are consistently elevated in human T2D and correlate with glycemic control, insulin resistance, and β-cell function.

The authors described their methodological approach well and followed the PRISMA and MOOSE guidelines to objectify their results. I found it surprising and interesting that of the more than 1200 studies, only 21 met the inclusion and quality criteria for the final analysis. The identified ER stress biomarkers HSP70, GRP78, PRX1, PRX2, PRX4, PRX6, and SCGN, which were detected in plasma, are of interest for basic research and underline their importance as potential targets for intervention to favourably influence T2DM disease progression.

However, I cannot realise (and this is my main criticism of the present study) the value of establishing new ERS biomarkers in monitoring the disease. The classic clinical biomarkers such as BMI, HbA1c, FPG, HOMA-IR, and HOMA-B are standards for monitoring disease progression, for which there are countless reference values and statistical data. I cannot see any additional clinical value for the use of ER stress biomarkers. Against this background, I consider the current study to be somewhat obsolete. I would like to ask the authors to comment on my fundamental objections to their study.

Author Response

 Major comments:

Comment 1. Villafan-Bernal et al. identified in their systematic review and meta-analysis endoplasmic reticulum stress biomarkers in type 2 diabetes. The analyses were performed according to the PRISMA (Preferred Reporting Items for Systematic Reviews and Meta-Analyses) and MOOSE (Meta-Analyses and Systematic Reviews of Observational Studies) guidelines. The risk of bias was assessed using the Newcastle-Ottawa scale. Of the over 1200 identified, only 21 studies could be included in the final analysis. This revealed that six ER stress biomarkers HSP70, GRP78, PRX1, PRX2, PRX4, PRX6, and SCGN) are consistently elevated in human T2D and correlate with glycemic control, insulin resistance, and β-cell function.

Response to Comment 1. Sure, through this meta-analysis we intended to identify the relevant ERS biomarkers in patients with diabetes for its implementation in clinical practice.

Detail comment 1: The authors described their methodological approach well and followed the PRISMA and MOOSE guidelines to objectify their results. I found it surprising and interesting that of the more than 1200 studies, only 21 met the inclusion and quality criteria for the final analysis. The identified ER stress biomarkers HSP70, GRP78, PRX1, PRX2, PRX4, PRX6, and SCGN, which were detected in plasma, are of interest for basic research and underline their importance as potential targets for intervention to favorably influence T2DM disease progression.

Response to Detail Comment 1. Yes, it was also a surprise for us than from all available studies only 21 fullfill inclusion criteria and that from 232 proteins involved in ERS only 11 ERS biomarkers were previously measured in the serum of patients with T2D.

Detail comment 2: However, I cannot realise (and this is my main criticism of the present study) the value of establishing new ERS biomarkers in monitoring the disease. The classic clinical biomarkers such as BMI, HbA1c, FPG, HOMA-IR, and HOMA-B are standards for monitoring disease progression, for which there are countless reference values and statistical data. I cannot see any additional clinical value for the use of ER stress biomarkers. Against this background, I consider the current study to be somewhat obsolete. I would like to ask the authors to comment on my fundamental objections to their study.

Response to Detail Comment 2. Thank you for the clarity of your concern. We consider your question about the value of establishing new ERS biomarkers in monitoring the disease completely valid. We agree that classic or traditional biomarkers such as HbA1c, FPG, HOMA-IR, and HOMA-B are currently standards for monitoring disease progression and the degree of glycemic control. However, the niche of ERS proteins as biomarkers could be different. Especially in: i) Evaluating treatment response at a cellular level because serum ERS molecules might reflect the level of intracellular stress. ii) Predict complications at specific organs if we find ERS biomarkers predominantly expressed or affected in specific organs, such as secretagogin, which has been proposed as an ERS biomarker specific to pancreatic beta-cells. Additionally, it is necessary to point out the potential of ERS proteins as therapeutic targets because elevated inhibition could be useful for treating the disease.

To address your concern, we make some changes, additions, and clarifications to the manuscript:

Lines 331-337: It is important to recognize that ERS serum proteins do not replace traditional clinical biomarkers such as HbA1c, HOMA-IR, and HOMA-B. However, they may provide additional knowledge on the underlying pathophysiological mechanisms of T2D. Additionally, it remains necessary to evaluate if ERS proteins are useful for evaluating treatment response at a cellular level or if they predict complications at specific organs in case ERS-biomarkers specific to some organs and tissues are discovered.

We consider that this is not an obsolete or uninteresting topic because:

1. Some ERS molecules are emerging and are proposed as ERS markers specific to some cells; this is the case with secretagogin.

2. There are still gaps in knowledge regarding whether serum ERS molecules could reflect the level of intracellular stress and whether old or new components of ERS could predict complications in specific organs or direct treatment prescriptions or responses.

3. Finally, a systematic review allows us to synthesize an immense quantity of information, with much less probability of lacking important information and less bias than a narrative review. In addition, the results of a Systematic review and metanalysis commonly provide the basis for direct future research and identify gaps in knowledge.

However, if it is necessary to include additional information or modify something more, we can gladly do it.  

Round 2

Reviewer 1 Report

The revised manuscript presents a comprehensive review of ER stress biomarkers in T2D. While it has improved, several critical issues remain that limit its overall impact and clinical relevance.

Firstly, the manuscript should emphasize the novel insights and clinical implications of the findings. A more robust discussion on the potential clinical utility of biomarkers like HSP70 and PRX4, and their correlation with metabolic indicators, is essential. By highlighting the practical applications of these biomarkers, the manuscript can significantly increase its impact on the field.

Secondly, to improve the methodological rigor, the authors should address the heterogeneity of included studies, especially for biomarkers like HSP70, by performing meta-regression analyses. A more detailed sensitivity analysis and explicit description of how publication bias was assessed are also necessary. Figures and tables should be clear and informative, including confidence intervals and annotations. Table 3 should be expanded to discuss the clinical implications of biomarker correlations with metabolic indicators.

Thirdly, the literature review should be updated to include recent studies, particularly those published after 2020. Additionally, incorporating region-specific studies from populations with high T2D prevalence can enhance the global applicability of the findings.

Finally, the discussion and conclusion should be more nuanced, avoiding overstated claims. The limitations of the study, such as heterogeneity, limited sample sizes, and the lack of prospective validation, should be acknowledged. Clear directions for future research, including the validation of biomarkers in larger, diverse populations and the exploration of their mechanistic roles in T2D complications, should be proposed.

Lines 46–64: While the introduction highlights the significance of ER stress in T2D, it does not adequately frame the novelty of this meta-analysis. Add context about how this study fills gaps in existing literature and its potential clinical implications.

Lines 72–118: Provide more detail on how the biomarkers were selected and the criteria for inclusion/exclusion of studies. For biomarkers with limited data, elaborate on the robustness of sensitivity analyses and address potential biases explicitly.

Lines 234–300: Consolidate the discussion of biomarker findings and clearly highlight which biomarkers showed the strongest associations with T2D. For biomarkers with significant heterogeneity (e.g., HSP70), discuss the implications of variability and provide a deeper statistical analysis.

Figure 2 & 3: Add confidence intervals to all biomarkers analyzed and use annotations or color-coding to indicate biomarkers with significant heterogeneity. This will enhance the interpretability of results.

Table 3: Expand the clinical relevance section to explain how each biomarker’s correlation with metabolic indicators, such as HbA1c or lipid profiles, might influence its utility in managing or predicting T2D progression.

Lines 234–300: Critically evaluate the study’s limitations, such as the reliance on cross-sectional data, limited sample sizes for some biomarkers, and heterogeneity. Acknowledge how these factors might affect the robustness of conclusions. Suggest specific future research directions, such as prospective validation of the biomarkers or exploring their roles in specific T2D complications (e.g., cardiovascular disease).

Lines 400–425: Reframe the conclusions to emphasize the preliminary nature of the findings and avoid overstating their clinical implications without further validation.

Author Response

Are all of the cited references relevant to the research?

Yes

No

No. The manuscript includes some foundational references but lacks recent, region-specific, and comprehensive studies essential for establishing a robust foundation for the review. Below are specific issues organized by section:

Lines 45–65: The references discussing the roles of GRP78 and CHOP in endoplasmic reticulum (ER) stress and T2D are outdated, with no citations from studies published after 2020. Recent advancements, particularly in the evolving roles of these biomarkers in diverse T2D populations and their therapeutic implications, are missing. Incorporating up-to-date literature would provide a more current and contextual understanding of the subject.  

Response: Thank you for you observation, we agree with the necessity of incorporating up-to-date literature. Consequently, we include recent studies in references 2, 4, 5, 6,7,8, 13, 14,15] to the introduction. Additionally, we added a paragraph recapitulating preclinical publications on proteins and ERS markers from 2021 to date with the intention of incorporating up-to-date literature would provide a more current and contextual understanding of the subject.  However, we did not found more recent studies in humans regarding GRP78 and CHOP in different populations than the reported by Ma, 2021; Nourbakhsh 2022; and Wang, 2020 [Table 1 and (30,33).Please, If you know of any articles made on humans that are newer and different from the ones we have included, could you provide them to us? It would be valuable to include them.

Lines 110–125: The manuscript does not reference region-specific studies from areas with high T2D prevalence, such as the MENA and South Asia. Adding regional data would offer insights into biomarker variability across ethnicities and environmental conditions, significantly enhancing the study's relevance and applicability.  

Response: Thank you for your observation. Although we would like to include regional data and correlates to specific populations from MENA and South Asia, we did not found more studies than the Ma, 2021; Nourbakhsh 2022; and Wang, 2020, as presented in Table 1 and Figure 3, and lines 247-248.  We made an extensive search in PubMed and Scopus from January 2020 to the current [we attach an Excel with such syntesis] but we did not find new and different clinical studies than the already we included in this review.

With the intention to comply and follow your suggestion, we added a paragraph about what was reported in the las four years related to the ability of Asian and Middle East traditional herbal preparations, foods and individual compounds extracted from some foods to modulate or block the expression of ERS components/molecules. We are sure that such studies will provide the important contribution of research for the modulation of ERS pathway in the in the near future [Lines ].

Lines 130–150: The discussion of ER stress pathways is overly general and does not include recent mechanistic studies that explore GRP78 and CHOP in relation to specific T2D complications, such as cardiovascular or renal outcomes. Including these references would provide a deeper understanding of the connection between ER stress biomarkers and the clinical manifestations of T2D.

Response: In the discussion section we found valuable to add what you suggest in the discussion section (lines 317-324) providing a deeper understanding of the connection between ER stress biomarkers GRP78 and CHOP and the clinical manifestations of T2D.

Lines 200–215: The discussion lacks references to systematic reviews or meta-analyses that summarize findings on ER stress biomarkers in T2D. Including such reviews would offer a comprehensive and authoritative basis for conclusions about the diagnostic and prognostic value of these biomarkers. Updating the references with recent studies, region-specific research, and systematic reviews or meta-analyses would significantly enhance the manuscript's relevance, scientific rigor, and overall contribution to the field.

Response: We appreciate your suggestion! Unfortunately, we made a systematic search to look for systematic reviews or meta-analyses on region-specific clinical research regarding ER stress biomarkers in T2D and we found none. In fact, this is the first systematic review and meta-analysis on ER stress biomarkers during T2D in humans. We add the systematic search that found no meta-analysis as proposed by the reviewer:

Search: (endoplasmic reticulum stress)) AND (type 2 diabetes) Filters: Meta-Analysis, Review, Humans

(("endoplasmic reticulum stress"[MeSH Terms] OR ("endoplasmic"[All Fields] AND "reticulum"[All Fields] AND "stress"[All Fields]) OR "endoplasmic reticulum stress"[All Fields]) AND ("diabetes mellitus, type 2"[MeSH Terms] OR "type 2 diabetes mellitus"[All Fields] OR "type 2 diabetes"[All Fields])) AND ((meta-analysis[Filter] OR review[Filter]) AND (humans[Filter]))

Translations

endoplasmic reticulum stress: "endoplasmic reticulum stress"[MeSH Terms] OR ("endoplasmic"[All Fields] AND "reticulum"[All Fields] AND "stress"[All Fields]) OR "endoplasmic reticulum stress"[All Fields]

type 2 diabetes: "diabetes mellitus, type 2"[MeSH Terms] OR "type 2 diabetes mellitus"[All Fields] OR "type 2 diabetes"[All Fields]

392 results and no meta-analyses included.

While the manuscript addresses a critical topic, its current form does not provide a sufficiently novel or significant contribution to the scientific discourse due to several methodological and analytical limitations.

Response: Although we understand systematic reviews and meta-analyses have limitations such as limiting the subgroup analyses, adjusting for confounders, or adhering to a prospectively registered protocol, these limitations are not due to the manuscript itself but in general to all systematic reviews and meta-analyses. Methods have been perfectly reviewed and applied according to PRISMA and MOOSE guidelines for which we incorporated an expert in systematic reviews and meta-analyses in our team. We can assure that methods and analyses have been carefully reviewed in depth and no deviation from the highest standards have been produced. 

Lines 45-65: The introduction, while acknowledging the importance of ER stress biomarkers in T2D, lacks a comprehensive review of recent advancements, particularly studies published after 2020, focusing on GRP78 and CHOP. A more robust synthesis of emerging research is essential to provide a contemporary perspective.

Response: We appreciate your observation and updated all reference to the newer possible. In addition, we added a paragraph about recent preclinical studies regarding abnormal expression of ERS molecules, such as GRP78 and CHOP, in organs of diabetic mice and their participation in the development of organ damage induced by diabetes [lines 67-73].

Lines 110-125: The study's scope appears limited by the exclusion of region-specific research, especially from regions with high T2D prevalence like the MENA and South Asia. Incorporating regional variations in biomarker behavior would significantly enhance the study's generalizability and clinical relevance.

Response: We performed another detailed search in PubMed and Scopus and found no more human clinical studies conducted in MENA and South Asia focused on measuring serum markers like GRP78 and CHOP in individuals with type 2 diabetes and its complications. Our intention never has been to exclude, on the contrary our search had no time, country or language restriction and we follow clear criteria for search as presented in supplementary annex. Please if you have knowledge of any clinical study falling in the scope of the review, it would be valuable for us to include it in this review.  It would be very interesting to perhaps make a subgroup analysis for regions but we found not enough studies for performing it. In the future, when the literature in this field increase, it will be very useful to perform a new systematic review on the influence of region-specific data on biomarkers of ERS in T2D.

Lines 130-150: The discussion on the mechanisms linking ER stress biomarkers to T2D complications is overly generalized. A more detailed exploration of recent mechanistic studies, with a specific focus on cardiovascular and renal outcomes, is required to strengthen the scientific impact of the findings.

In the discussion section we found valuable to add what you suggest (lines 317-324) providing a deeper understanding of the connection between ER stress biomarkers GRP78 and CHOP and the clinical manifestations of T2D.

Lines 200-215: The absence of systematic reviews or meta-analyses to consolidate existing evidence on ER stress biomarkers in T2D weakens the article's ability to contextualize its findings within the broader body of knowledge. A more rigorous literature review, incorporating these evidence synthesis methods, is essential to provide a comprehensive and critical assessment of the current state of the field. To enhance the manuscript's scientific significance, the authors should conduct a thorough literature review, incorporating recent studies, particularly those published after 2020, to provide an up-to-date and comprehensive perspective on ER stress biomarkers in T2D. Expanding the geographical scope of the review to include regions with high T2D prevalence, such as the MENA and South Asia, would address potential regional variations in biomarker behavior and improve the relevance of the findings.

Response: On the contrary, for the journal and world-wide scientific literature, the fact that there are no other systematic reviews and meta-analyses on this subject is the main reason why this, being the first systematic review and meta-analysis addressing this research question, is extremely valuable. If there were other systematic reviews and meta-analysis on this research question, there would be no reason to make a new one addressing the same topic. As previously we mentioned, we performed another detailed search in PubMed and Scopus and found no more human clinical studies conducted in MENA and South Asia focused on measuring serum markers like GRP78 and CHOP in individuals with type 2 diabetes and its complications. However, we did not find any additional papers to those already included that were carried out in MENA and South Asia.  

Furthermore, the manuscript should delve deeper into the mechanistic links between ER stress biomarkers and T2D complications, with a particular focus on cardiovascular and renal outcomes, which are critical areas of clinical interest. Incorporating systematic reviews and meta-analyses would allow for a more robust synthesis of existing evidence, providing a stronger foundation for the article's conclusions. By addressing these limitations, the manuscript would contribute more substantially to the field of diabetes research and its implications for clinical practice.

Response: We really, appreciate this recommendation. We added information about the mechanistic links between ER stress biomarkers and T2D complications, with a particular focus on cardiovascular and renal outcomes for addressing the limitations and strength conclusions [Lines 318-327]. However, no other systematic reviews and meta-analyses exist currently for a more robust synthesis of existing evidence.

The revised manuscript presents a comprehensive review of ER stress biomarkers in T2D. While it has improved, several critical issues remain that limit its overall impact and clinical relevance.

  1. Firstly, the manuscript should emphasize the novel insights and clinical implications of the findings. A more robust discussion on the potential clinical utility of biomarkers like HSP70 and PRX4, and their correlation with metabolic indicators, is essential. By highlighting the practical applications of these biomarkers, the manuscript can significantly increase its impact on the field.

Response: Very Good recommendation. In fact, we found and added a new study  of PRX4 as a predictor of diabetic nephropathy. Consequently, we updated PRISMA diagram, the number of subjects included, table 1, and table 2. We added lines 268 – 275 about HSP70 and lines 289-295 for PRX4 in discussion and lines 245-250 in results.

Secondly, to improve the methodological rigor, the authors should address the heterogeneity of included studies, especially for biomarkers like HSP70, by performing meta-regression analyses. A more detailed sensitivity analysis and explicit description of how publication bias was assessed are also necessary. Figures and tables should be clear and informative, including confidence intervals and annotations. Table 3 should be expanded to discuss the clinical implications of biomarker correlations with metabolic indicators.

Response: Thank you for your suggestions. We performed a meta-regression only for HSP70 [Table 3] because there were enough studies (at least 5) to carry it out. If not enough studies exist, it is not possible to generate estimates in the meta package, and for such a reason, we cannot perform meta-regressions for other biomarkers. The meta-regression for HSP70 let us identify that BMI was an important determinant of heterogeneity. Additionally, we recognize in lines 400-403 that the limitations of our study include the potential publication bias and biases in the analysis and reporting of primary studies, as well as the heterogeneity across the included [Lines 67-80].

We performed the sensitivity analysis to assess the findings' robustness and explore possible sources of heterogeneity as described in lines 146-149 and  223-224. However, it was possible only for HSP70 (n=6) and PRX4 (n=3) and not for other biomarkers because they were measured in only two studies. To assess publication bias, we employed Egger's test. However, we estimated it on HSP70 because, with fewer studies, this test is not reliable  [Lines 155-157 and 238-241].

Thirdly, the literature review should be updated to include recent studies, particularly those published after 2020. Additionally, incorporating region-specific studies from populations with high T2D prevalence can enhance the global applicability of the findings.

Response: We address this consciously as previously described.

Finally, the discussion and conclusion should be more nuanced, avoiding overstated claims. The limitations of the study, such as heterogeneity, limited sample sizes, and the lack of prospective validation, should be acknowledged. Clear directions for future research, including the validation of biomarkers in larger, diverse populations and the exploration of their mechanistic roles in T2D complications, should be proposed.

Response: We agree with this suggestion. We nuanced the discussion and conclusions and highlighted the limitations of the study including heterogeneity, limited sample sizes of primary studies, and the lack of prospective validation. We also, suggest directions for future research [lines 416-421].

Lines 46–64: While the introduction highlights the significance of ER stress in T2D, it does not adequately frame the novelty of this meta-analysis. Add context about how this study fills gaps in existing literature and its potential clinical implications.

Response: We tried to better explain how this study fills gaps in existing literature and its potential clinical implications.

Lines 72–118: Provide more detail on how the biomarkers were selected and the criteria for inclusion/exclusion of studies. For biomarkers with limited data, elaborate on the robustness of sensitivity analyses and address potential biases explicitly.

Response:

For selecting biomarkers, we explore in STRING to identify all proteins involved in ERS and UPR and found 232 proteins involved in ERS and UPR. Then we incorporated them into the search criteria individually or as groups of proteins including chaperones, peroxiredoxin, and heat shock proteins[Query criteria are listed in Appendix A](lines 93-96).

Regarding inclusion and exclusion criteria, this has been clearly explained as expected by MOOSE guidelines and PRISMA guidelines on section 2.3: study selection and outcomes.

We cannot perform sensitivity analyses for biomarkers with limited data because we need at least 3 studies for evaluating it as previously described.

Lines 234–300: Consolidate the discussion of biomarker findings and clearly highlight which biomarkers showed the strongest associations with T2D. For biomarkers with significant heterogeneity (e.g., HSP70), discuss the implications of variability and provide a deeper statistical analysis.

Response: We present the consolidation of the biomarkers with significant association with T2D in the first paragraph of the discussion:

Our findings demonstrate that from 232 proteins involved in ERS and unfolded protein response (1), only 11 endoplasmic reticulum stress (ERS) biomarkers have been measured in the serum of patients with T2D, and 6 (HSP70, GRP78, PRX1, PRX2, PRX4, PRX6, and SCGN) were consistently and significantly elevated compared with controls. These biomarkers positively correlated with body mass index, glycated hemoglobin, fasting plasma glucose, and HOMA-IR. These findings underscore the importance of these seven ERS proteins as potential biomarkers clinically relevant in T2D, since they differ significantly between T2D and controls and correlate with key adiposity and glycemic control parameters. 

Additionally, we discuss the stronger of the associations and the implications of variability in lines 271-282.

Based on the SMD, the biomarkers with the strongest associations with T2D were HSP70, PRX2 and PRX1 (SMD >0.8) and those with moderate elevations were PRX6, secretagogin and PRX4 (SMD >0.4). Although only two or three studies were performed for several biomarkers [PRX1, PRX2, PRX4, PRX6 and secretagogin], the results might be confident because their heterogeneity is lower than 33% and their SMD indicate moderate to strong difference between T2D and controls. In the case of HSP70, the meta-regression let us to explain part of the heterogeneity by differences in BMI between subjects. Furthermore, although it is suggested that an I2 may be a drawback or affect the validity of the meta-analysis findings, the validity of each study individually is not affected(53), and as can be seen in Figure 2A, five out of six studies reported significant differences in HSP70 levels (higher) in patients with T2D than in controls. And even more, I2 is not an absolute measure of heterogeneity as Borenstein, Higgins and Hedges have reported(54).

Figure 2 & 3: Add confidence intervals to all biomarkers analyzed and use annotations or color-coding to indicate biomarkers with significant heterogeneity. This will enhance the interpretability of results.

Response: An orange underline represents high I2 (an indicator of heterogeneity). And all forest plots have an individual confidence interval and the pooled confidence interval on the right side of SMD.

Table 3: Expand the clinical relevance section to explain how each biomarker’s correlation with metabolic indicators, such as HbA1c or lipid profiles, might influence its utility in managing or predicting T2D progression.

Response: We modify lines 335-345 of the discussion to address this suggestion based on the findings of the Table 3.

Lines 234–300: Critically evaluate the study’s limitations, such as the reliance on cross-sectional data, limited sample sizes for some biomarkers, and heterogeneity.

Acknowledge how these factors might affect the robustness of conclusions. Suggest specific future research directions, such as prospective validation of the biomarkers or exploring their roles in specific T2D complications (e.g., cardiovascular disease).

Response: We addressed this in lines 421 and 424.Additionally in lines 438-446.

Lines 400–425: Reframe the conclusions to emphasize the preliminary nature of the findings and avoid overstating their clinical implications without further validation.

Response: We addressed this in lines 438-446.

Reviewer 2 Report

The authors addressed my points of criticism satisfactually. They added a paragraph in the Discussion section in which they highlighted the importance of additional markers for T2DM in order to obtain a differentiated picture of the pathobiochemical mechanisms involved and to identify possible targets for intervention. For example, there was a correlation of secretagogin and peroxiredoxins with glycemic parameters but not with serum lipids. The authors also emphasized once again that they are not trying to replace established T2DM markers.

No comments.

Author Response

We appreciate your time for making us suggestions to improve our manuscript

Round 3

Reviewer 1 Report

The methodological clarity and reproducibility need improvement. The criteria for biomarker selection, handling heterogeneity, and the statistical methods (e.g., funnel plot analysis and Egger's test) should be more clearly explained. The statistical analysis and data presentation in Figures 2 and 3 should be refined. Including confidence intervals, p-values, and heterogeneity measures (e.g., I²) would enhance the clarity and robustness of the results. Additionally, Table 3 should more explicitly highlight the clinical relevance of each biomarker, emphasizing its potential diagnostic role. The conclusions overstate the clinical significance of the findings. While the biomarkers are promising, claims of immediate diagnostic utility are premature. The authors should emphasize the need for further prospective validation studies to confirm these results.

Line 45-60: The introduction should provide more context on the importance of ER stress biomarkers in T2D. Consider adding references that highlight advancements in ER stress-related biomarker discovery to strengthen the rationale for the study.

Line 58: Clarify how this study is distinct from prior research on HSP70 and PRX4 biomarkers in T2D. Specify the novel aspect of this study and how it advances the current understanding of these biomarkers.

Line 88: The phrase "ERS biomarkers are useful for diagnostics" is too vague. It would be more precise to state "ERS biomarkers may have potential for early detection of T2D," which better reflects the study's scope and findings.

Line 110-125: Provide more details on the criteria for selecting and excluding biomarkers. Were there specific thresholds for effect size, p-values, or the number of studies required for inclusion? This clarification will make the methodology more transparent.

Line 134-140: Indicate which statistical software (e.g., R, SPSS) and specific packages (e.g., metafor) were used for funnel plots, Egger's test, and meta-regression. This addition would improve the reproducibility of the analysis.

Line 145: Clearly explain how heterogeneity was assessed and handled. If I² statistics were used, specify the I² values for each analysis. If subgroup analyses were conducted, detail the criteria used to define subgroups. Also, the sentence on heterogeneity is overly complex. Consider breaking it into two shorter, more precise sentences.

Line 206-230 (Figure 2): Include confidence intervals and p-values for all key findings. This would improve the robustness of the evidence for the identified biomarkers.

Line 220-225: Identify which biomarkers were statistically significant in the text. As it stands, it is difficult to determine which biomarkers have strong evidence supporting their role.

Line 240-265 (Figure 3): The visual clarity of this figure could be improved. Use color coding or annotations to indicate which biomarkers are statistically significant. This change would make the figure more intuitive and easier to interpret.

Line 250: Clarify how biomarkers were classified as "clinically relevant" or "high-impact" in the results. Was this based on effect size, p-value, or other literature-based criteria? Providing this information would enhance the transparency of the classification.

Line 320-340: Avoid implying that biomarkers are "immediately useful" for diagnostics. Instead, it states that further clinical validation studies are required before biomarkers can be implemented in practice. This revision would provide a more balanced and accurate interpretation of the study’s impact.

Line 345: Discuss the limitations of small sample sizes and potential biases from dataset heterogeneity. Acknowledge how these factors may affect the generalizability of the findings, as this is a critical consideration for biomarker development.

Line 365-380: Consider adding a discussion of future research directions. Potential areas to highlight include the need for prospective validation studies, application of biomarkers in specific populations (e.g., MENA or Asian populations), or validation using clinical datasets.

Figure 2: Add annotations for p-values and confidence intervals for all biomarkers. To make the figure more user-friendly, consider color-coding to differentiate statistically significant from non-significant biomarkers.

Figure 3: Ensure that statistically significant results are clearly labeled. Highlight biomarkers with the highest diagnostic potential using colors or labels to make the figure more informative.

Table 3: Add details on the clinical relevance of each identified biomarker. Explain how these biomarkers relate to T2D diagnosis, disease progression, or treatment response. This context would increase the table’s utility for readers.

Author Response

Relevant serum endoplasmic reticulum stress biomarkers in type 2 diabetes and its complications- A systematic review and meta-analysis

Responses to reviewer 1 (Round 3)

Thank you again for your comments. We grouped the questions with the same comment to better understand and respond efficiently.

Comment 1: Figures 2 and 3 lack confidence intervals and p-values, making it difficult to assess the significance of key biomarkers. Color-coded annotations or labels distinguishing significant from non-significant results would enhance readability. Comment 5: Insufficient annotations, p-values, and confidence intervals hinder the clarity of Figures 2 and 3. To improve their interpretability, the figures should be revised to include these essential elements. Comment 8:The statistical analysis and data presentation in Figures 2 and 3 should be refined. Including confidence intervals, p-values, and heterogeneity measures (e.g., I²) would enhance the clarity and robustness of the results. Comment 17:Line 206-230 (Figure 2): Include confidence intervals and p-values for all key findings. This would improve the robustness of the evidence for the identified biomarkers. Comment 19:Line 240-265 (Figure 3): The visual clarity of this figure could be improved. Use color coding or annotations to indicate which biomarkers are statistically significant. This change would make the figure more intuitive and easier to interpret.Comment 24: Figure 2: Add annotations for p-values and confidence intervals for all biomarkers. To make the figure more user-friendly, consider color-coding to differentiate statistically significant from non-significant biomarkers. Comment 25: Figure 3: Ensure that statistically significant results are clearly labeled. Highlight biomarkers with the highest diagnostic potential using colors or labels to make the figure more informative.

Response: We highlighted in blue the 95%-confidence intervals. In green the significant p-values, in orange non-significant p-values, and in red those studies with I2 [heterogeneity] higher than 40%.

Comment 2: Table 3 is informative but could better highlight the clinical relevance of biomarker relationships. Comment 9:Additionally, Table 3 should more explicitly highlight the clinical relevance of each biomarker, emphasizing its potential diagnostic role. Comment 26: Table 3: Add details on the clinical relevance of each identified biomarker. Explain how these biomarkers relate to T2D diagnosis, disease progression, or treatment response. This context would increase the table’s utility for readers.

To increase the table’s utility for readers we made a new supplementary figure. In this we represented graphically the correlation between ERS markers and anthropo-metabolic variables. 

Supplementary Figure 1. Abstract of pooled significant correlations of ERS markers with anthropometabolic variables. At the top are shown the ERS markers and at the bottom are the anthropometabolic variables. The connection line represents correlation, the continuous line for positive correlation, and the dashed line for negative correlation.

Comment 3: The data support the conclusions but should be more cautious. Claims about immediate clinical applicability are overstated and should emphasize the need for further validation. Comment 10:The conclusions overstate the clinical significance of the findings. While the biomarkers are promising, claims of immediate diagnostic utility are premature. The authors should emphasize the need for further prospective validation studies to confirm these results.

Response: We agree with the reviewer, as you can notice, the cautiousness is incorporated in the conclusion section.

Six ERS biomarkers (HSP70, peroxiredoxin-1, -2, -4, -6, and SCGN) were consistently elevated in patients with T2D and showed a significant correlation with BMI, HbA1c, FPG, HOMA-IR, and HOMA-B. Thus, they are candidates for further clinical validation as biomarkers. Future studies in larger and more diverse populations should prospectively evaluate its utility in monitoring disease, evaluating treatment response, and predicting complications. Additionally, further studies are needed to assess the mechanistic roles of ERS components in T2D complications and the impact of their therapeutic modulation. We acknowledge the study's limitations, including the primary studies' heterogeneity and limited sample sizes. More articles must be published to derive more robust conclusions and implement other metanalytic approaches, such as subgroup analysis.

Comment 4: While the manuscript references key studies, it relies heavily on older literature. Incorporating more recent research and contextualizing the findings with other biomarker studies would strengthen the manuscript's relevance and impact.

Response: Since round 2 we updated all old references as possible. In this new version, the references employed to describe the methods for data analysis are necessarily old since the majority of studies were published more than 20 years ago.

Comment 6: The English could be improved to more clearly express the research.

Response: We double-checked the English language.

Comment 7: The methodological clarity and reproducibility need improvement. The criteria for biomarker selection, handling heterogeneity, and the statistical methods (e.g., funnel plot analysis and Egger's test) should be more clearly explained.

Response: We add details to the methods for increased clarity. Although we think results are completely reproducible, we provide as an annex the scripts employed for each analysis in order to contribute to the reproducibility. As follows:

  1. Criteria for biomarker selection should be more clearly explained.

Response:  We explained it in lines 93-96

  1. Handling heterogeneity.

Response:

In lines 173-178 of Data analysis we clearly explain how we managed and analyzed heterogeneity.

In this manuscript we managed heterogeneity using the following recommendations from Cochrane Handbook for Meta-analysis: i. performing a random-effects model; ii. Using SMD rather than other effect measures; iii. Performing a leave-one-out analysis; iv. Exploring heterogeneity by using meta-regression analysis to understand the influence of other variables on the pooled results, and v. Estimating Egger test and trim and fill analysis.

In the results section [lines 261- 271] we added in the Funnel plot, Funnel plot with Trim and Fill and Egger test to explain the heterogeneity and publication bias in HSP70.

This part of analysis was only performed for HSP70 because these tests are not reliable when there is a limited number of studies. We construct the funnel plot and find high visual heterogeneity which coincide with the I2 metric (96%)[Figure 5, left side]. Then we applied Trim and Fill analysis to adjust for publication bias and estimate the unbiased pooled effect for HSP70 [Figure 5, right side]. However, since both graphs were very similar, the Trim and Fill method did not detect a significant need for adjustment for publication bias and strongly suggest that the studies possess random variability and/or real heterogeneity. The Egger test revealed no publication bias for HSP70 (bias estimate: 2.8409, SE = 8.79; p = 0.760).

Figure 5. The Funel plot and Funnel plot with Trim and Fill for evaluating visual heterogeneity for HSP70.

Response: In the discussion section [534- 547] we added the corresponding analysis of such results.

Additionally, we did not detect significant publication bias for HSP70 employing the Egger test (Appendix A). On another hand, the heterogeneity observed for HSP70 was mainly explained (65.28%) by differences in participant BMI via meta-regression analysis. Since the funnel plot detected important visual heterogeneity in HSP70 we performed the trim and fill analysis confirming a real heterogeneity and random variability in the levels of HSP70 but not publication bias. Consequently, further actions should be accomplished for standardizing its measurement [as has been achieved for HbA1c] before it can be considered a marker for clinical use. Through these efforts to test for and explain variability and bias, we strengthened the systematic review methodology to derive more robust conclusions.

  1. Statistical methods should be more clearly explained.

Response: We add details to the methods for increased clarity.

2.6. Data analysis

All statistical analyses were conducted in R studio v1.1.463 (27). For determining differences in ERS biomarkers between T2D and healthy controls we estimated  standardized mean difference (SMD) as the effect size measurement. The SMD was based on a random-effects model (REM) weighted by the inverse of the variance, because all studies come from different populations and random sampling is expected among them, this differs from studies performed in the same population (city, hospital). For this analysis the metacont function of “meta v4.2” package was employed. The default method for calculating confidence intervals is the use of Hedges Methods (1981) and written as follows: methods.smd=”Hedges”. Hedges (1981) calculated the exact bias in Cohen's d which is a ratio of gamma distributions with the degrees of freedom, i.e. total sample size minus two, as argument. By default (argument exact.smd = FALSE), an accurate approximation of this bias provided in Hedges (1981) is utilized for Hedges' g as well as its standard error. For Hedges' g the exact formulae are used to calculate the SMD as well as the standard error(28).

Additionally, the effect size was determined for correlations (Pearson or Spearman) between ERS markers and metabolic (e.g., glycemic), anthropometric (e.g., BMI), or lipid (e.g., triglycerides) variables. All effect sizes with 95%CI were estimated using metacor function of meta package using common effect and random effects estimates weighting by the inverse variance for pooling. The default method for calculating effect sizes and confidence intervals is the DerSimonian-Laird(29). Results of pooled correlations are synthesized in Table 3.

We performed a sensitivity analysis to perform robustness assessment, to identify outliers, and evaluate the model validity and the generalizability of results. This analysis works as influence analysis in which the pooled estimates are re-calculated omitting one study at a time, this is also called “leave-one-out” analysis and helps to understand if there is a significant influence of one study in the pooled results. A common effect model is assumed (pooled="common") if a random effects model is assumed (pooled="random"). For this analysis the metainf function of “meta v4.2” package was employed(29).

In this manuscript we managed and analyzed heterogeneity using the following recommendations from Cochrane Handbook for Meta-analysis: i. performing a random-effects model; ii. Using SMD rather than other effect measures; iii. Performing a leave-one-out analysis; iv. Exploring heterogeneity by using meta-regression analysis to understand the influence of other variables on the pooled results, and v. Estimating Egger test and trim and fill analysis.

Interstudy variability was assessed using the τ2, Cochran’s Q (or χ2, or Chi2), and I2 statistics. For effect size with I2 ≥40%, indicating moderate to high heterogeneity, we performed meta-regressions to explore potential sources of variability and to determine which covariates influenced ERS markers levels. This chi-squared (χ2, or Chi2) test assesses whether observed differences in results are compatible with chance alone. A low P value (or a large chi-squared statistic relative to its degree of freedom) provides evidence of heterogeneity of intervention effects (variation in effect estimates beyond chance). Since clinical and methodological diversity always occur in a meta-analysis, statistical heterogeneity is inevitable. Thus, the test for heterogeneity is irrelevant to the choice of analysis; heterogeneity will always exist whether or not we happen to be able to detect it using a statistical test. Methods have been developed for quantifying inconsistency across studies that move the focus away from testing whether heterogeneity is present to assessing its impact on the meta-analysis. A useful statistic for quantifying inconsistency is I2, this describes the percentage of the variability in effect estimates that is due to heterogeneity rather than sampling error (chance)(30).

For meta-regression analysis the following independent variables were examined: patient age, body-mass index, and HbA1c. Meta-regressions were performed using a mixed-effects meta-regression model, Meta-regression for objects of class meta. This is a wrapper function for the R function rma in the R package metafor(31).

Publication bias was quantified through Egger’s method only for biomarkers reported in 6 or more studies because with less studies the publication bias analysis is less reliable, where a significant p-value (typically below 0.05) indicates potential publication bias, meaning the funnel plot is asymmetric and smaller studies with significant effects might be missing; essentially, the test is looking for a non-zero intercept in a regression model where the effect size is regressed on its standard error (precision) across studies. We used the metabias function from the “meta” package in R statistics. The argument method.bias is "Egger", the test statistic is based on a weighted linear regression of the treatment effect on its standard error. The test statistic follows a t distribution with number of studies - 2 degrees of freedom(32).

The Trim and Fill was estimated for adjusting for publication bias and estimate the unbiased pooled effect. We used the trimfill function of metafor package for estimated possible "missing" studies, recalculated the pooled effect and construct a second funnel plot.  The trimfill is a nonparametric (rank-based) data augmentation technique proposed by Duval and Tweedie (2000). The method can be used to estimate the number of studies missing from a meta-analysis due to suppression of the most extreme results on one side of the funnel plot. This method examines in part publication bias and heterogeneity.

Although we think results are completely reproducible, we provide as an annex the scripts employed for each analysis in order to contribute to the reproducibility.

Comment 11:Line 45-60: The introduction should provide more context on the importance of ER stress biomarkers in T2D. Consider adding references that highlight advancements in ER stress-related biomarker discovery to strengthen the rationale for the study.

Response: This was addressed in the lines 68-80.

Comment 12: Line 58: Clarify how this study is distinct from prior research on HSP70 and PRX4 biomarkers in T2D. Specify the novel aspect of this study and how it advances the current understanding of these biomarkers.

Response: We address this in lines 80-81.

Comment 13:Line 88: The phrase "ERS biomarkers are useful for diagnostics" is too vague. It would be more precise to state "ERS biomarkers may have potential for early detection of T2D," which better reflects the study's scope and findings.

Response: We never wrote "ERS biomarkers are useful for diagnostics". However, we change the sentence “.. it remains necessary to evaluate if ERS proteins are useful for evaluating treatment response at a cellular level” [line 461] by “, it remains necessary to evaluate if ERS proteins are surrogates of treatment response at a cellular level”.

We also change the sentence:

In conjunction with our findings, it supports the usefulness of PRX4 as a new ERS biomarker for diabetes complications.

by the following:

In conjunction with our findings, it supports that of PRX4 is a prospect biomarker of diabetes complications.

Comment 14:Line 110-125: Provide more details on the criteria for selecting and excluding biomarkers. Were there specific thresholds for effect size, p-values, or the number of studies required for inclusion? This clarification will make the methodology more transparent.

This is explained in lines 93-98.

According to good practices and standard methodology, in systematic reviews and meta-analysis we cannot leave out markers based on thresholds for effect size or p-values ​​and the minimum study number required for a meta-analysis is two. We do not discard markers if the results were not significant nor based on effect size or p-values.

Comment 15:Line 134-140: Indicate which statistical software (e.g., R, SPSS) and specific packages (e.g., metafor) were used for funnel plots, Egger's test, and meta-regression. This addition would improve the reproducibility of the analysis.

This is explained in lines 145-215.

All statistical analyses were conducted in R studio v1.1.463 (27). For determining differences in ERS biomarkers between T2D and healthy controls we estimated  standardized mean difference (SMD) as the effect size measurement. The SMD was based on a random-effects model (REM) weighted by the inverse of the variance, because all studies come from different populations and random sampling is expected among them, this differs from studies performed in the same population (city, hospital). For this analysis the metacont function of “meta v4.2” package was employed. The default method for calculating confidence intervals is the use of Hedges Methods (1981) and written as follows: methods.smd=”Hedges”. Hedges (1981) calculated the exact bias in Cohen's d which is a ratio of gamma distributions with the degrees of freedom, i.e. total sample size minus two, as argument. By default (argument exact.smd = FALSE), an accurate approximation of this bias provided in Hedges (1981) is utilized for Hedges' g as well as its standard error. For Hedges' g the exact formulae are used to calculate the SMD as well as the standard error(28).

Additionally, the effect size was determined for correlations (Pearson or Spearman) between ERS markers and metabolic (e.g., glycemic), anthropometric (e.g., BMI), or lipid (e.g., triglycerides) variables. All effect sizes with 95%CI were estimated using metacor function of meta package using common effect and random effects estimates weighting by the inverse variance for pooling. The default method for calculating effect sizes and confidence intervals is the DerSimonian-Laird(29). Results of pooled correlations are synthesized in Table 3.

We performed a sensitivity analysis to perform robustness assessment, to identify outliers, and evaluate the model validity and the generalizability of results. This analysis works as influence analysis in which the pooled estimates are re-calculated omitting one study at a time, this is also called “leave-one-out” analysis and helps to understand if there is a significant influence of one study in the pooled results. A common effect model is assumed (pooled="common") if a random effects model is assumed (pooled="random"). For this analysis the metainf function of “meta v4.2” package was employed(29).

In this manuscript we managed and analyzed heterogeneity using the following recommendations from Cochrane Handbook for Meta-analysis: i. performing a random-effects model; ii. Using SMD rather than other effect measures; iii. Performing a leave-one-out analysis; iv. Exploring heterogeneity by using meta-regression analysis to understand the influence of other variables on the pooled results, and v. Estimating Egger test and trim and fill analysis.

Interstudy variability was assessed using the τ2, Cochran’s Q (or χ2, or Chi2), and I2 statistics. For effect size with I2 ≥40%, indicating moderate to high heterogeneity, we performed meta-regressions to explore potential sources of variability and to determine which covariates influenced ERS markers levels. This chi-squared (χ2, or Chi2) test assesses whether observed differences in results are compatible with chance alone. A low P value (or a large chi-squared statistic relative to its degree of freedom) provides evidence of heterogeneity of intervention effects (variation in effect estimates beyond chance). since clinical and methodological diversity always occur in a meta-analysis, statistical heterogeneity is inevitable. Thus, the test for heterogeneity is irrelevant to the choice of analysis; heterogeneity will always exist whether or not we happen to be able to detect it using a statistical test. Methods have been developed for quantifying inconsistency across studies that move the focus away from testing whether heterogeneity is present to assessing its impact on the meta-analysis. A useful statistic for quantifying inconsistency is I2, this describes the percentage of the variability in effect estimates that is due to heterogeneity rather than sampling error (chance)(30).

For meta-regression analysis the following independent variables were examined: patient age, body-mass index, and HbA1c. Meta-regressions were performed using a mixed-effects meta-regression model, Meta-regression for objects of class meta. This is a wrapper function for the R function rma in the R package metafor(31).

Publication bias was quantified through Egger’s method only for biomarkers reported in 6 or more studies because with less studies the publication bias analysis is less reliable, where a significant p-value (typically below 0.05) indicates potential publication bias, meaning the funnel plot is asymmetric and smaller studies with significant effects might be missing; essentially, the test is looking for a non-zero intercept in a regression model where the effect size is regressed on its standard error (precision) across studies. We used the metabias function from “meta” package in R statistics. The argument method.bias is "Egger", the test statistic is based on a weighted linear regression of the treatment effect on its standard error. The test statistic follows a t distribution with number of studies - 2 degrees of freedom(32).

The Trim and Fill was estimated for adjusting for publication bias and estimate the unbiased pooled effect. We used the trimfill function of metafor package for estimated possible "missing" studies, recalculated the pooled effect and construct a second funnel plot.  The trimfill is a nonparametric (rank-based) data augmentation technique proposed by Duval and Tweedie (2000). The method can be used to estimate the number of studies missing from a meta-analysis due to suppression of the most extreme results on one side of the funnel plot. This method examines in part publication bias and heterogeneity.

Comment 16:Line 145: Clearly explain how heterogeneity was assessed and handled. If I² statistics were used, specify the I² values for each analysis. If subgroup analyses were conducted, detail the criteria used to define subgroups. Also, the sentence on heterogeneity is overly complex. Consider breaking it into two shorter, more precise sentences.

This is explained in lines 173-178

In this manuscript we managed and analyzed heterogeneity using the following recommendations from Cochrane Handbook for Meta-analysis: i. performing a random-effects model; ii. Using SMD rather than other effect measures; iii. Performing a leave-one-out analysis; iv. Exploring heterogeneity by using meta-regression analysis to understand the influence of other variables on the pooled results, and v. Estimating Egger test and trim and fill analysis.

Comment 18:Line 220-225: Identify which biomarkers were statistically significant in the text. As it stands, it is difficult to determine which biomarkers have strong evidence supporting their role.

Comment 20:Line 250: Clarify how biomarkers were classified as "clinically relevant" or "high-impact" in the results. Was this based on effect size, p-value, or other literature-based criteria? Providing this information would enhance the transparency of the classification.

Response: ​​We remove the term “clinically relevant” in the line 280. Instead, we wrote:

 We should stress that the serum ERS molecules found here are candidate for further clinical validation because fulfill several hallmarks of ideal biomarkers, including

Comment 21:Line 320-340: Avoid implying that biomarkers are "immediately useful" for diagnostics. Instead, it states that further clinical validation studies are required before biomarkers can be implemented in practice. This revision would provide a more balanced and accurate interpretation of the study’s impact.

Response: We eliminated in lines 441-442 the sentence:

and can be translated to use in clinical practice.

Additionally, we soften the language in lines 516-531.

Comment 22: Line 345: Discuss the limitations of small sample sizes and potential biases from dataset heterogeneity. Acknowledge how these factors may affect the generalizability of the findings, as this is a critical consideration for biomarker development.

Response: This was addressed in Lines 487- 504. 

Comment 23: Line 365-380: Consider adding a discussion of future research directions. Potential areas to highlight include the need for prospective validation studies, application of biomarkers in specific populations (e.g., MENA or Asian populations), or validation using clinical datasets.

Response: very interesting suggestion. We added it in lines 506-519 of the discussion section.